

# Quantifying the volatility of organic aerosol in the southeastern U.S.

Provat K. Saha[1], Andrey Khlystov[2], KhairunnisaYahya[3], Yang Zhang[3], Lu Xu[4], Nga L. Ng[4,5] , and Andrew P. Grieshop[1]

[1]Department of Civil, Construction and Environmental Engineering, North Carolina State University, Raleigh, NC, USA
[2]Division of Atmospheric Sciences, Desert Research Institute, Reno, Nevada, USA
[3]Department of Marine Earth and Atmospheric Sciences, North Carolina State University, Raleigh, NC, USA
[4]School of Chemical and Biomolecular Engineering, Georgia Institute of Technology, Atlanta, GA, USA
[5]School of Earth and Atmospheric Sciences, Georgia Institute of Technology, Atlanta, GA, USA

*Correspondence to*: Andrew P. Grieshop (apgriesh@ncsu.edu)

**Abstract.** The volatility of organic aerosols (OA) has emerged as a property of primary importance in understanding their atmospheric lifecycle, and thus abundance and transport. However, quantitative estimates of the thermodynamic (volatility) and kinetic parameters dictating ambient OA gas-particle partitioning, such as saturation concentrations ($C^*$), enthalpy of evaporation ($\Delta H_{vap}$) and evaporation coefficient ($\gamma_e$), are highly uncertain. Here, we present measurements of ambient OA volatility at two sites in the southeastern U.S., one at biogenic-volatile-organic-compound (BVOC)-dominated rural setting in Alabama as part of the Southern Oxidant and Aerosol Study (SOAS) in June-July, 2013, and another at a more anthropogenically-influenced urban location in North Carolina during October-November, 2013. These measurements applied a dual-thermodenuder (TD) system, in which temperature and residence times are varied in parallel, to constrain equilibrium and kinetic aerosol volatility properties. Gas-particle partitioning parameters were determined via evaporation kinetic model fits to the dual-TD observations. OA volatility parameters values derived from both datasets were similar despite the fact that measurements were collected in distinct settings and seasons. The OA volatility distributions also did not vary dramatically over the campaign period nor strongly correlate with OA components identified via positive matrix factorization of aerosol mass spectrometer data. A large portion (40-70%) of measured ambient OA at both sites was composed of very low volatility organics ($C^* \leq 0.1\ \mu g\ m^{-3}$). An effective $\Delta H_{vap}$ of bulk OA of $\sim$ 80-100 kJ mol$^{-1}$ and a $\gamma_e$ value of $\sim$ 0.5 best describe the evaporation observed in the TDs. This range of $\Delta H_{vap}$ values is substantially higher than that typically assumed for simulating OA in atmospheric models (30-40 kJ mol$^{-1}$). TD data indicate that $\gamma_e$ is on the order of 0.1 to 0.5, indicating that repartitioning timescales for atmospheric OA are on the order of several minutes to an hour under atmospheric conditions. The OA volatility distributions resulting from fits were compared to those simulated in the Weather, Research and Forecasting model with Chemistry (WRF/Chem) with a current treatment of SOA formation. The substantial fraction of low-volatility material observed in our measurements is largely missing from simulations, and OA mass concentrations are underestimated. The large discrepancies between simulations and observations indicate a need to treat low volatility OA in atmospheric models. Volatility parameters extracted from ambient measurements enable evaluation of emerging treatments for OA (e.g., secondary OA using the volatility basis set or formed via aqueous chemistry) in atmospheric models.



## 1. Introduction

Organic aerosol (OA) is a dominant component of atmospheric fine particulate matter (PM$_{2.5}$) (Jimenez et al., 2009; Zhang et al., 2007), which is linked with adverse human health and uncertain climate effects. Atmospheric OA is a complex mixture of thousands of individual organic compounds originating from a range of natural and anthropogenic sources. Primary OA (POA)

is emitted directly into the atmosphere whereas secondary OA (SOA) is formed in the atmosphere via condensation of low-volatility products of oxidation reactions of volatile organic compounds (VOCs). A large fraction of SOA in many parts of the globe, e.g., in the southeast U.S., is formed from biogenic-VOCs (BVOCs) (Goldstein et al., 2009; Goldstein and Galbally, 2007). However, the mechanisms responsible for SOA production from BVOCs (Budisulistiorini et al., 2015; Goldstein and Galbally, 2007, 2007; Marais et al., 2016; Xu et al., 2015a, 2015b), its chemical composition and many important physical

properties are largely undetermined (Goldstein et al., 2009; Schichtel et al., 2008; Weber et al., 2007). Therefore their representation in current atmospheric and climate models are highly uncertain (Hallquist et al., 2009; Liao et al., 2007; Pye et al., 2015; Pye and Seinfeld, 2010).

One of the major sources of uncertainty in predicting SOA concentrations in atmospheric models arises from the poor understanding of gas-particle partitioning of chemical species comprising SOA (Hallquist et al., 2009; Jimenez et al., 2009;

Seinfeld and Pankow, 2003). Gas-particle partitioning, dictated by the parameters that define OA volatility, plays a central role in determining OA lifecycle and thus its atmospheric abundance, transport, and impacts (Donahue et al., 2006; Jimenez et al., 2009). At equilibrium, saturation vapor pressure (or equivalently, saturation concentration, $C^*$; μg m$^{-3}$) determines whether an organic compound is found in the particle- or gas-phase (Donahue et al., 2006; Pankow, 1994) and enthalpies of vaporization ($\Delta H_{vap}$) dictate the change in partitioning with temperature (Epstein et al., 2010). Although gas-particle partitioning is

determined by the basic thermodynamic properties of OA species – their $C^*$ and $\Delta H_{vap}$ (Donahue et al., 2006, 2012) – these, along with the impacts of non-ideal mixing on individual species, are generally unknown for ambient OA. Under changing conditions, gas-particle partitioning is also influenced by the kinetics of gas/particle exchange, for example due to barriers to mass transfer in solid or viscous particles or molecular accommodation at a particle surface (Kroll and Seinfeld, 2008). The overall kinetic limitation to mass transfer during repartitioning is typically described by an evaporation coefficient ($\gamma_e$) (also

often called mass accommodation coefficient), which is highly uncertain for ambient OA and can dictate time-scales for partitioning (Saleh et al., 2013). Though current models assume OA to be at equilibrium within a model prediction time-step (several minutes to an hour) during atmospheric simulations, several studies have indicated that partitioning time scales could be as long as days or months ($\gamma_e \ll 0.1$) due to a highly viscous and/or glassy aerosol (Vaden et al., 2011; Zobrist et al., 2008).

Quantitative measures of ambient OA gas-particle partitioning parameters ($C^*$, $\Delta H_{vap}$, $\gamma_e$) are needed to provide inputs

for, and to evaluate, atmospheric models. However, methods to quantitatively determine ambient OA volatility are in their infancy and the resulting estimates of parameters dictating OA volatility are highly uncertain (Cappa and Jimenez, 2010). Thermodenuder (TD) systems have been previously applied to measure ambient OA volatility (Burtscher et al., 2001; Huffman et al., 2009; Lee et al., 2010; Paciga et al., 2015; Xu et al., 2016). A TD system measures evaporation of sampled aerosol at



various temperature perturbations by systematically comparing the size distribution and/or aerosol mass concentration measured after heating in a TD and at a reference ("bypass") condition (Huffman et al., 2008). Several efforts have been made to infer ambient OA volatility distributions by fitting observed evaporation in a TD using a model of evaporation kinetics (Cappa and Jimenez, 2010; Lee et al., 2010). However, since OA evaporation in a TD is dictated by a large number of independent parameters (e.g., $C^*$, $\Delta H_{vap}$, and $\gamma_e$) (Cappa and Jimenez, 2010; Lee et al., 2010), it is difficult to constrain all parameters with a single-dimensional perturbation (e.g. varying TD temperature) to the initial equilibrium. Saha et al. (2015) showed that operating two TDs in parallel (dual-TD) that vary both temperature and residence time can provide tighter constraint on estimates of volatility parameter values ($C^*$, $\Delta H_{vap}$, and $\gamma_e$) for single component OA via kinetic model fits to the observations. In Saha and Grieshop (2016), this approach was applied to determine volatility and phase-partitioning parameter values for laboratory α-pinene SOA. The resulting parameters are consistent with recent observations of low-volatility SOA (Jokinen et al., 2015; Zhang et al., 2015) and evaporation rates (Vaden et al., 2011; Wilson et al., 2015) observed by several techniques.

This paper describes the application of the dual-TD approach during ambient observations from two different settings in the southeastern U.S. Measurements at a rural site during the Southern Oxidant and Aerosol Study (SOAS-2013) (https://soas2013.rutgers.edu/) leverage the range of complementary measurements available during this large field study. To provide a contrast, measurements were also taken several months later, under cooler conditions, in Raleigh, U.S., a small metropolitan area in a similar ecological zone, but with stronger influence from local anthropogenic emissions. The objectives of the study were to: (i) determine a set of volatility parameter values, such as OA volatility distribution using the Volatility Basis Set (VBS) framework (Donahue et al., 2006, 2012) and $\Delta H_{vap}$ and $\gamma_e$, that describe observations, (ii) examine the variability and consistency in ambient OA volatility distributions across diverse settings and conditions, (iii) examine relationships between extracted volatility distributions and OA composition and source contributions, and (iv) evaluate a model treatment of OA volatility by comparing the measured OA volatility distribution with that simulated by a chemical transport model using a current implementation of the VBS framework.

## 2. Methods

### 2.1. Measurement sites

Ambient OA volatility measurements were conducted at two locations in the southeastern U.S., one in a forested rural setting and another in an urban location. Six weeks (June 1 to July 15, 2013) of continuous measurements were conducted in rural Alabama during the Southern Oxidant and Aerosol Study (SOAS-2013) field campaign. The SOAS field campaign occurred in summer 2013 at several locations in the southeastern U.S. in order to study the interaction of biogenic and anthropogenic atmospheric compounds with a focus on BVOCs and organic aerosols. The measurements reported here are from the main SOAS ground site (32.903°N, 87.250°W), near Talladega National Forest and Centreville, Alabama. The Centreville, Alabama



site is an ideal location to study volatility of OA dominated by secondary OA from BVOC precursors (Warneke et al., 2010) in the presence of a range of anthropogenic influences. An additional four weeks (October 18 to November 20, 2013) of ambient OA volatility measurements were conducted at the North Carolina State University (NCSU) main campus (35.786°N, 78.669°W) in Raleigh, USA. The NCSU site, while in an area with plentiful tree cover and BVOC emissions, receives a substantially stronger influence from anthropogenic emissions due to its location within the Raleigh metro area. Section 3.1 includes further comparison between two study areas. Here-in-after, the two data sets are referred to as 'Centreville' and 'Raleigh'.

## 2.2. Dual thermodenuder operation and sampling strategy

Measurements were collected using the dual TD experimental setup introduced in Saha et al. (2015) and only briefly described here. Two TDs operated in parallel, one at various temperature settings (temperature stepping TD; TS-TD) with a fixed, relatively longer residence time (Rt) and another at fixed temperature and various Rt settings (variable residence time TD; VRT-TD). The TS-TD temperature settings were 40, 60, 90, 120, 150, and 180 ℃ with ~50 s Rt, while the VRT-TD operated at 60 or 90℃ with Rt varying between 1 to 40 s (5-8 settings). All Rts reported here are calculated assuming plug flow at room temperature. Temperature effects on Rt were included during modeling of evaporation kinetics (discussed below) as $Rt(T_{TD})$ = $Rt(T_{ref}) \times (T_{ref}/T_{TD})$, where $T_{ref}$ and $T_{TD}$ are the reference (e.g., room temperature) and TD temperature in K, respectively (Cappa, 2010). The time to run through all temperatures and Rt steps during measurements was ~ 4-5 hours.

A schematic of the experimental setup is shown in Fig.1. Three Scanning Mobility Particle Sizers (SMPS, TSI Inc; Model 3081 DMAs; Model 3010/3787 CPCs) simultaneously measured aerosol size distributions (10-600 nm) in 3 parallel lines (two TDs and one bypass). An Aerosol Chemical Speciation Monitor (ACSM, Aerodyne Research Inc.) alternated between the bypass and TS-TD lines at ~ 20-30 minute intervals using an automated 3-way valve system. The ACSM measured the sub-micron aerosol (~ 75-650 nm) mass concentration of non-refractory chemical species (organic, sulfate, nitrate, ammonium, and chloride) (Ng et al., 2011a).

All aerosol instruments and TD inlets were inside a temperature-controlled (25˚±2) trailer in Centreville, and laboratory room in Raleigh. Ambient air was continuously sampled through a sampling inlet located on the rooftop of a trailer/building (~5 m above ground level). The sampling inlet included a $PM_{2.5}$ cyclone (URG Corp, 16.7 L min$^{-1}$) followed by a ~ 8 mm inner diameter copper sample line. A silica gel diffusion dryer upstream of TD inlets and aerosol instruments maintained relative humidity (RH) < 30-40%.

## 2.3. Quantifying OA evaporation

Evaporation of bulk OA at a particular TD operating temperature and residence time is described in terms of mass fraction remaining (MFR). OA MFR is the ratio of OA mass concentration measured after passing through TD to that measured via the bypass (room temperature) line. For quantitative assessment of aerosol volatility, such as during modeling of aerosol





evaporation, the initial OA concentration ($C_{OA}$) and particle size are also needed. Empirically estimated particle loss correction factors as a function of TD temperatures and residence times (Saha et al., 2015) and instrumental inter-calibration factors were applied in MFR calculations. Since the VRT-TD line was measured with the SMPS only (Fig.1), it provided only information on evaporation of submicron aerosol in terms of its volume concentration. We estimated the OA MFR from VRT-TD/SMPS

data assuming measured aerosol volume was comprised of OA and ammonium sulfate (AS) only. This is a reasonable assumption under these conditions because more than 90% of measured aerosol volume concentrations can be explained by OA+AS for both sites (see supplementary information; SI, Fig. S1). Our calculations also assumed that AS did not evaporate at the VRT-TD operating temperatures (60 or 90˚C) (Fig. S2). For further detail on the estimation of approximate OA MFR from VRT-TD/SMPS data, see SI, section S.1.

**2.4. Determining OA phase partitioning parameters**

We apply a previously described volatility parameter extraction framework (Saha et al., 2015; Saha and Grieshop, 2016) to extract a set of volatility parameter ($C^*$, $\Delta H_{vap}$, $\gamma_e$) values via inversion of dual-TD data using an evaporation kinetics model. The approach is outlined briefly below. The resulting fit describes OA using a $\log_{10}$ volatility basis set (VBS) framework (Donahue et al., 2006, 2012), where material is lumped into volatility bins separated by orders of magnitude in $C^*$ space at a

reference temperature ($T_{ref}$; 298 K). The volatility distribution extracted using this approach is an empirical estimate describing the bulk volatility behavior of OA, assuming absorptive partitioning and ideal mixing (unity activity coefficient) (Donahue et al., 2006, 2012). In the VBS approach, total OA concentration ($C_{OA}$; µg m⁻³) is modeled using Eq. 1.

$$C_{OA} = C_{tot} \sum_i f_i \left( 1 + \frac{C_i^*}{C_{OA}} \right)^{-1} \tag{1}$$

Here, $C_{tot}$ is the total organic material (vapor + aerosol) in phase equilibrium with $C_{OA}$; $f_i$ is the fraction of $C_{tot}$ in each volatility ($\log_{10} C^*$) bin. Thus, $f_{i =} C_{tot,i}/C_{tot}$, describes the distribution of organics in volatility space and is usually called the 'volatility

distribution'.

The Clausius-Clapyeron equation (Eq. 2) is used to represent temperature dependent $C^*$.

$$C_i^*(T) = C_i^*(T_{ref}) \exp\left[ -\frac{\Delta H_{vap,i}}{R} \left( \frac{1}{T} - \frac{1}{T_{ref}} \right) \right] \frac{T_{ref}}{T} \tag{2}$$

Where R is the gas constant and $\Delta H_{vap}$ is the enthalpy of vaporization.

To extract the volatility distribution of OA from ambient measurements, we select lower and upper $C^*$ ($T_{ref}$) bins of $10^{-4}$ and $10^1$ µg m⁻³, respectively. The selection of the lower and upper bins are determined by the highest TD operating temperature

(180 ˚C) and the average ambient OA loading ($C_{OA}$ ~ 5 µg m⁻³), respectively. With the above $C^*$ bin limits, materials having $C^* < 10^{-4}$ µg m⁻³ are lumped into the lowest bin, while materials having $C^* > 10$ µg m⁻³ are not represented.  Note, if a $C^*$ bin



of 100 µg m⁻³ is included, Eq. 1 indicates less than 5% of the material in this bin will be in the condensed-phase at $C_{OA}$ ~ 5 µg m⁻³. Therefore, $C^*$ bins > 10 µg m⁻³ are not well constrained by our TD data and are not included in our analysis.

The general approach to fitting a volatility parameterization employed in this study is similar to that applied to laboratory aerosol systems (Saha et al., 2015; Saha and Grieshop, 2016). Briefly, the kinetic model tracks both particle- and gas-phase concentrations of model species (each represented by a VBS bin) as they proceed through TD operated at a particular temperature and residence time. The model takes inputs of several aerosol properties (e.g., $C^*$ distribution, $\Delta H_{vap}$, diffusion coefficient ($D$), surface tension ($\sigma$), molecular weight ($MW$) and density ($\rho$)), total aerosol loading ($C_{OA}$) and particle diameter ($d_p$) and determines how much aerosol mass concentration will evaporate for a set of input parameters at a particular TD temperature and residence time. Non-continuum effects on mass transfer are represented using the Fuchs−Sutugin correction factor, which depends on $\gamma_e$. The model is applied in an inverse sense to extract OA properties such as the volatility distribution, $\Delta H_{vap}$, and $\gamma_e$ as fitting parameters by matching measured and modeled evaporation data. Values of D, $\sigma$, MW, and $\rho$ generally have a smaller influence on observed evaporation (Cappa and Jimenez, 2010; Saha et al., 2015), and are approximated from literature values (Table S-1). Volume median diameter was used as a representative $d_p$. For simplicity, a large ($\Delta H_{vap}$, $\gamma_e$) space was considered for fitting a $f_i$ distribution of measured OA. Following previous work (Epstein et al., 2010; May et al., 2013), a linear relationship was assumed between $\Delta H_{vap}$ and $\log_{10}C^*$ with $\Delta H_{vap,i}$ = intercept-slope ($\log_{10}C^*_{i,298}$), where intercept and slope are fit parameters. Values for $\Delta H_{vap}$ intercept = [50, 80, 100, 130, 200] and slope = [0, 4, 8, 11] KJ mol⁻¹ were applied along with $\gamma_e$ = [0.01, 0.05, 0.1, 0.25, 0.5, 1]. $\gamma_e$ was assumed constant over all bins, and is an effective parameter representing all kinetic limitations within the condensed-phase and at the particle surface.

A distribution of $f_i$ was solved for each combination of ($\Delta H_{vap}$, $\gamma_e$) applying the non-linear constrained optimization solver 'fmincon' in Matlab (Mathworks, Inc.) by first fitting TS-TD data; 'accepted' solutions were then further refined by fitting VRT-TD observations. A constraint of $\Sigma f_i$ =1 was used. The goodness of fit was quantified in terms of the sum of squared residual (SSR) values. For the campaign average fit, an 'acceptance' threshold value for SSR was selected based on observed variability (±one standard deviation) in measurements. A parameter set ($f_i$, $\Delta H_{vap}$, and $\gamma_e$) was considered a finally 'accepted' solution if it optimally reproduced both TS-TD and VRT-TD observations within the observed variability. The 'best fit' is defined as that with the lowest SSR value among all the accepted combinations.

## 2.5 Simulation of OA in a chemical transport model

Considering that VBS-based parameterizations are becoming common means to enhance the simulation of OA in chemical transport models (CTMs) (Farina et al., 2010; Lane et al., 2008b; Matsui et al., 2014; Shrivastava et al., 2013), measurements of OA volatility provide a useful means by which to evaluate these simulations. We compared OA volatility distributions measured in this study to those resulting from CTM simulations with a current VBS-based parameterization implemented in a modified version of the Weather, Research and Forecasting model with Chemistry (WRF/Chem) v3.6.1 (Wang et al., 2015; Yahya et al., 2016b). The WRF/Chem simulation uses the Carbon Bond version 6 (CB6) gas-phase mechanism (Yarwood et



al., 2010) coupled by Wang et al. (2015) to the Model for Aerosol Dynamics for Europe – Volatility Basis Set (MADE/VBS) (Ackermann et al., 1998; Ahmadov et al., 2012; Shrivastava et al., 2011). The CB6-MADE/VBS treatment includes semivolatile POA and SOA, as well as a fragmentation and functionalization treatment for multi-generational OA aging based on Shrivastava et al.(2013). The fragmentation and functionalization treatment in this case assumes 25% fragmentation for the third and higher generations of oxidation (Shrivastava et al., 2013). The ranges of $C^*$ values used in WRF/Chem  simulation are defined based on current SOA and semi-volatile POA parametrizations and were $10^0$ to $10^3$ μg m$^{-3}$ for ASOA (anthropogenic-SOA) and BSOA (biogenic-SOA), $10^{-2}$ to $10^6$ μg m$^{-3}$ for POA and $10^{-2}$ to $10^5$ μg m$^{-3}$ for SVOA (semi-volatile OA), where SVOA refers to oxidized OA from evaporated POA. The semi-empirical correlation for $\Delta H_{vap}$ by Epstein et al. (2010) was used to estimate temperature-dependent partitioning.

The simulations are performed at a horizontal resolution of 36-km with $148 \times 112$ horizontal grid cells over the continental U.S. domain and parts of Canada and Mexico, and a vertical resolution of 34 layers from the surface to 100-hPa. Anthropogenic emissions in 2010 are based on the 2008 National Emissions Inventory (NEI) from the Air Quality Model Evaluation International Initiative (AQMEII) project (Pouliot et al., 2015). Biogenic emissions are simulated online by the Model of Emissions of Gases and Aerosols from Nature v2.1 (MEGAN2.1) (Guenther et al., 2012). The chemical initial and boundary conditions (ICs/BCs) come from the modified Community Earth System Model/ Community Atmosphere Model (CESM/CAM v5.3) with updates by He and Zhang (2014) and Gantt et al.(2014). The meteorological ICs/BCs come from National Center for Environmental Protection Final Analysis (FNL) data.

## 3. Results

### 3. 1. Overview of campaign characteristics

The two field campaigns were conducted in settings with distinct local emission sources and metrological conditions. The Centreville campaign was during summer (T=24.7 ± 3.3˚C, RH = 83.1 ± 15.3%). Local organic emissions surrounding the Centreville site are dominated by BVOCs since this site is located in a forest and biogenic emissions substantially increase with temperature (Lappalainen et al., 2009; Tarvainen et al., 2005; Warneke et al., 2010). In contrast, Raleigh measurements were in a setting with substantially stronger anthropogenic emissions during fall/winter (T=12.7 ± 6.0˚C, RH = 65.7 ± 18.8%). Comparison of long-term data from an air quality monitoring station near the Raleigh site shows substantially higher $NO_x$ (5-10 fold) and CO (2-4 fold) concentrations relative to those observed at Centreville (See Fig. S3). However, the Raleigh-Durham metropolitan area has plentiful tree cover and thus substantial local BVOC emissions. For instance, α- and β-pinene concentrations measured in summer at Centreville and Duke Forest (about 40 km Northwest of the Raleigh site) are in the same range (Fig. S4). However, since the Raleigh campaign was conducted at lower temperature conditions, local BVOC emissions are expected to be lower by a factor of 3 to 4 (Fig S.4). Measurements in such diverse but similar ecological settings allows us to examine the consistency of OA volatility under varying levels of biogenic and anthropogenic contribution.



Figs. S5 to S7 show average meteorological conditions, submicron aerosol size distributions, chemical composition and their temporal variations over the campaign periods. Ambient submicron particle number concentrations (10-600 nm) were higher in Raleigh (Centreville: 1500-3000 cm$^{-3}$, Raleigh: 3000-6000 cm$^{-3}$) and particle size was relatively smaller (volume median diameter, Centreville: 275 ± 30 nm, Raleigh: 227 ± 34 nm) (Fig. S6). Organic species were the dominant component

in non-refractory submicron aerosol (PM$_1$) as measured by the ACSM at both sites (Centreville: 71± 10 %, Raleigh: 76 ± 8 %). The campaign average ± one standard deviation of ACSM-derived OA mass concentrations were 5.2 ± 3.0 μg m$^{-3}$ in Centreville and 6.7 ± 3.6 μg m$^{-3}$ in the Raleigh campaign, assuming a collection efficiency (CE) of 0.5. Application of the 'coarse' tracer $m/z$ based factor analysis approach to decompose OA mass spectra (Ng et al., 2011b), the majority of OA measured at both sites was oxygenated-OA (OOA). While approximately 7% of campaign averaged OA mass concentration

in Raleigh was classified as hydrocarbon-like OA (HOA), the HOA contribution at the Centreville site was negligible. Positive matrix factorization (PMF) results from high-resolution mass spectra collected at the Centerville site (Xu et al., 2015a, 2015b) and their linkage with the measured OA volatility are discussed in sections 3.3 and 3.4, below.

### 3. 2. Observed campaign average evaporation of OA

Fig.2 shows the campaign average OA MFR as a function of TD temperature and residence time. (1-MFR) at a TD temperature

and residence time indicates what fraction of bulk OA mass evaporates at that condition. It is important to note that MFR at a given temperature is not a consistent descriptor of OA volatility because it depends on many parameters related to TD experimental conditions (e.g., $Rt$) and sampled aerosol (e.g., $C_{OA}$, $d_p$). Therefore, MFR data should not be interpreted as a direct measure of OA volatility or even directly compared (unless experiments are conducted under identical conditions).

Fig. 2a (MFR vs. temperature, frequently called a thermogram plot) shows TS-TD measurements from this study along

with one other measurement from SOAS (Hu et al., 2016) and several previous field and laboratory measurements. The campaign average OA MFRs measured at the two sites in the southeastern US, under relatively consistent $C_{OA} \sim 5$ μg m$^{-3}$, were found to be quite similar. Approximately 60 -70% of OA mass evaporated after heating at 100℃ with a residence time of 50 s. The campaign average T$_{50}$ and T$_{90}$ (temperature at which 50% and 90% of OA mass evaporate, respectively) with a residence time of 50 s were ~ 78℃ and ~ 180℃, respectively. Data from α-pinene chamber SOA experiments collected using the same

dual-TD setup at atmospheric conditions (dark ozonolysis, $C_{OA} \sim 5$ μg m$^{-3}$), described in Saha and Grieshop (2016), are also shown. Relative to the ambient observations, the lab SOA data show similar evaporation behavior in the lower temperature range (40-90 ˚C), but relatively greater evaporation at higher temperatures.

Fig. 2b and 2c show the campaign-average estimated OA MFRs at various residence times with the VRT-TD operated at 60℃ and 90℃, respectively. Results show increased evaporation with longer residence time. In Fig. 2a, data are color coded

by the TD residence time used in each study. A substantial effect of residence time on the observed evaporation is consistent with that observed across TD measurements from several previous field studies. This effect of residence time on observed MFR strongly suggests that comparisons of OA volatility across studies should not be made based on measured MFRs. Doing





so may bias inferences about differences in aerosol volatility. Observed evaporation depends on TD residence time and many physical and chemical properties of sampled aerosol (Cappa, 2010; Riipinen et al., 2010; Saleh et al., 2011), unless the aerosol reaches equilibrium inside a TD (saturates the gas phase across the volatility range). The equilibration time of aerosol in a TD is dictated by many parameters including particle size distribution, diffusion coefficient (D) and evaporation coefficient ($\gamma_e$) and is typically several minutes or more under atmospheric (low $C_{OA}$) conditions (Saleh et al., 2011, 2013).

Following the method of Saleh et al. (2013), the estimated characteristic equilibration times for the sampled aerosol in the Centreville and Raleigh measurements are 147-470 s and 150-450 s, respectively, assuming unhindered mass transfer ($\gamma_e = 1$). These calculations are based on the interquartile ranges of particle number concentrations ($N_p$) and condensation sink diameter ($d_{cs}$) measured in Centreville ($N_p \sim$ 1500-3000 cm$^{-3}$, $d_{cs} \sim$ 125- 170 nm) and Raleigh ($N_p \sim$ 3000-6000 cm$^{-3}$, $d_{cs} \sim$ 80- 105 nm), $D = 3.5 \times 10^{-6}$ m$^2$ s$^{-1}$ and $MW = 200$ g mol$^{-1}$. A factor-of-ten reduction in $\gamma_e$ relative to ideal accommodation ($\gamma_e = 0.1$) increases equilibration time by an order of magnitude. The observed continuous downward slope of MFR versus residence time (Fig. 2b, c) suggests that equilibrium was not reached in the TD during the maximum Rt of 50 s. This result implies that TD measurements in an ambient setting are essentially a measure of the evaporation rate of sampled aerosol, rather than one of volatility/$C^*$, an equilibrium thermodynamic property. Therefore, an evaporation kinetic model is needed to extract volatility parameter values from ambient TD data.

**3.3. Extracted OA volatility parameter values**

Fig. 3 presents the results of the extraction process used to determine parameters dictating gas-particle partitioning ($f_i$, $\Delta H_{vap}$, $\gamma_e$); the example shown is for a fit to the Centreville campaign-average data, though the same process was conducted for all fits. Fig. S8 shows a similar plot for the Raleigh data set. Fitting results show that a broad range of $\gamma_e$ (0.05 to 1) can reproduce the TS-TD observation within observed variability (i.e., error bars in Fig. 2) for several $\Delta H_{vap}$ combinations (accepted TS-TD fits are shown with filled inner circles). The inclusion of VRT-TD data provides additional constraints for parameter fitting. Only the points with white crosses (x) in Fig. 3 recreate both TD data sets; a larger sized 'x' represents a better fit. Thus, VRT-TD data help to narrow the possible solution space. Fig. 3 shows that $\Delta H_{vap} = 100$ KJ mol$^{-1}$ and $\gamma_e = 0.5$ provide the overall *best fit* for the Centreville data set. For the Raleigh data set, $\Delta H_{vap}$ of both 80 (marginally better) and 100 KJ mol$^{-1}$ with $\gamma_e =$ 0.5 provide similarly good fits (Fig S8). For simplicity, $\Delta H_{vap} = 100$ KJ mol$^{-1}$ and $\gamma_e = 0.5$ are considered as best estimates for both data sets for the next portion of the paper.

These results are inconsistent with a very small value of OA evaporation coefficient (e.g., $\gamma_e \ll 0.1$) that would indicate significant resistance to mass transfer during evaporation, which has been suggested previously based on dilution (Grieshop et al., 2007, 2009; Vaden et al., 2011) and heating (Lee et al., 2011) experiments. Our best estimate of $\gamma_e \sim 0.5$ is consistent with the observations of Saleh et al. (2012), in which they report an $\gamma_e \sim$ 0.28 to 0.46 for ambient aerosols in Beirut, Lebanon via measured equilibrium profiles of concentrated ambient aerosols ($C_{OA} \sim$ 200-300 μg m$^{-3}$) after heating in a TD at 60 ˚C. Our results show that an effective $\gamma_e \sim$ 0.1 to 1 can explain dual-TD data to within observed variability, suggesting that there



is no extreme resistance to mass transfer such as what might be encountered due to a glassy-solid or highly-viscous aerosol. Some previous assertions of highly inhibited evaporation (Grieshop et al., 2007; Vaden et al., 2011) were likely biased as they assumed volatility distributions based on smog-chamber yield experiments that likely overestimated the volatility, and thus expected evaporation rate of lab OA (Saha and Grieshop, 2016; Saleh et al., 2013).

Our fitting results show that a $\Delta H_{vap}$ intercept of 80 -130 KJ mol$^{-1}$ and slopes of 0 or 4 KJ mol$^{-1}$ can be used to explain campaign average observations (Fig. 3, Fig. S6). These $\Delta H_{vap}$ values are consistent with those of atmospherically-relevant low-volatility organics such as dicarboxylic acids (Bilde et al., 2015), but distinct from those typically assumed (30 - 40 KJ mol$^{-1}$) for atmospheric modeling (Farina et al., 2010; Lane et al., 2008b; Pye and Seinfeld, 2010). The semi-empirical correlation based fit from Epstein et al.(2010) ($\Delta H_{vap}=130-11 \log_{10}C^*$) has steeper $\log_{10}C^*$ dependence than those able to explain our
observations (Figs. 3, S8).

Although several $\Delta H_{vap}$ and $\gamma_e$ combinations can recreate observations from both TDs within variability (Figs. 3, S8), to enable comparison of $C^*$ distributions we adopt our *best* estimates of $\Delta H_{vap}$ and $\gamma_e$ ($\Delta H_{vap} = 100$ KJ mol$^1$ and $\gamma_e = 0.5$) for further analysis of data from both campaigns. Campaign-average $f_i$ distributions corresponding to that ($\Delta H_{vap}$, $\gamma_e$) are the basis for model fits shown in Fig. 2. The 'campaign-average' $f_i$ distribution was derived by fitting campaign-average dual-TD
observations (Fig. 2) and using campaign-average $C_{OA}$ and $d_p$. A $f_i$ distribution was also fit based on all the individual measurements from the campaign (*MFR, $C_{OA}$, $d_p$*; 20-30 minute time resolution) using $\Delta H_{vap} = 100$ KJ mol$^{-1}$ and $\gamma_e = 0.5$; we term this the 'unified' fit. The 'campaign-average' and 'unified' $f_i$ distributions for the Centreville and Raleigh data set are listed in Table 1. In addition to the volatility distribution ($f_i$), we also show estimates of mean $C^*$ ($\overline{C^*}$; estimated as $\overline{C^*} = 10^{\sum f_i \log_{10} C_i^*}$) to quantify the center of mass (central tendency) of different volatility distributions. Another way to collapse a
distribution to a single value (also reported in Table 1) is the effective $C^*$ ($C_{eff}^*$) of the ensemble, estimated as, $C_{eff}^* = \sum x_i C_i^*$, where $x_i$ is the condensed-phase mass fraction in each $C_i^*$ bin and $\sum x_i = 1$. While the volatility parameter values reported in Table 1 are the *best* fit results, other parameter sets can reproduce observations within variability. Application of $f_i$ distributions reported in Table 1 must be with reported $\gamma_e$ and $\Delta H_{vap}$ values. Sensitivities of the estimated volatility parameter values to assumed values of D, $\sigma$, *MW,* and $\rho$ are discussed in Saha et al.(2015) and Saha and Grieshop(2016). These assumed parameters
have relatively minor effects on observed evaporation in a TD compared to $C^*$, $\gamma_e$, and $\Delta H_{vap}$.

The extracted campaign-average and unified-fit OA volatility distributions ($f_i$) and corresponding $\overline{C^*}$ and $C_{eff}^*$ from Centreville and Raleigh data sets are quite similar (see Table 1). A large portion of the measured OA (40-70%) at both sites is composed of very low-volatility organics (LVOCs; $C^* \leq 0.1$ µg m$^{-3}$, Donahue et al., 2012). It is somewhat surprising that results from two field campaigns, which occurred in distinct scenarios with varying level of biogenic and anthropogenic emissions,
results in such similar OA volatility distributions. This finding is consistent with those of Kolesar et al.(2015a), who report similar mass thermograms for laboratory SOA formed from a variety of anthropogenic and biogenic VOCs under different oxidant (O$_3$, OH) conditions. Our extracted ambient OA volatility distributions are also comparable to those previously derived





from TD measurement in Mexico City (Cappa and Jimenez, 2010) and Finokalia, Greece (Lee et al., 2010). However, the ambient OA volatility distributions determined here are relatively less volatile than those from chamber-generated fresh SOA from α-pinene ozonolysis (Table 1).

Fig. 4a demonstrates a forward modeling exercise to show how the extracted average volatility parameter values ($f_i$, $\Delta H_{vap}$, and $\gamma_e$; those listed in Table 1) can reproduce individual measurements from the whole Centreville campaign as well as TD data from other groups (Cerully et al., 2015; Hu et al., 2016) during SOAS. The results show that a single set of volatility parameter values (campaign average/unified fit $f_i$, $\gamma_e$ = 0.5 and $\Delta H_{vap}$ = 100 KJ mol$^{-1}$) reproduce individual observations from the whole campaign within approximately ± 20%. These parameter values also closely reproduced the measured campaign average OA MFRs from the University of Colorado TD (Rt ~ 15 s) (Hu et al., 2016) and Georgia Tech TD (Rt ~ 7 s) (Cerully et al., 2015) collected during the Centreville campaign (see solid blue squares and cyan triangles in Fig. 4a). MFRs reported in Cerully et al. (2015) are for the total submicron aerosol species. These were converted to OA MFRs applying the method given in SI Sec. S1 to enable direct comparison with modeled OA MFRs.

Fig 4b shows a comparison of the extracted campaign-average OA volatility distribution from this study with those from two other independent approaches during the Centreville campaign (Hu et al., 2016; Lopez-Hilfiker et al., 2016). Hu et al. (2016) report OA volatility distributions from observed evaporation in a TD during the Centreville campaign fit using the method given by Faulhaber et al.(2009). In this method, TD evaporation observations at different temperatures are translated to a volatility distribution using an empirically derived calibration curve based on evaporation of known compounds and their $C^*$ (Faulhaber et al., 2009). Our derived distribution from dual-TD observations coupled with evaporation kinetic model is comparable to that from Hu et al. (2016), although this distribution is slightly less volatile than ours. Lopez-Hilfiker et al. (2016) derived an OA volatility distribution from Centreville measurements with the Filter Inlet for Gases and AEROsols-Chemical Ionization Mass Spectrometer (FIGAERO-CIMS), which thermally desorbs filter-bound aerosol into a CIMS. The FIGAERO-derived distribution is several orders of magnitude less volatile than ours; all OA in it has $C^* \leq 10^{-4}$ µg m$^{-3}$. Therefore, in Fig.4b the Centreville campaign-average $C_{OA}$ of ~ 5 µg m$^{-3}$ is assigned to the log10$C^* \leq$ -4 bin to enable direct comparison with TD-ACSM/AMS measurements (this study and Hu et al.). However, in reality FIGAERO-CIMS observations accounted for ~50% of AMS organic mass concentrations measured at Centreville (Lopez-Hilfiker et al. 2016), indicating half the OA was not quantified. The discrepancy between FIGAERO- and TD-based distributions would be reduced if this unmeasured OA is distributed in higher volatility bins, thus re-assigning material shown in the lowest volatility bin in Fig. 4b. Lopez-Hilfiker et al. (2016) reported that heating OA at higher temperatures has the potential to introduce artifacts into quantification of its volatility, for example if it causes oligomer decomposition leading to artificially high volatility. If this occurs, this may bias any heating-based measurement approaches, including TD measurements.

A test for these various parameter values is to use them to recreate data from other (non-heating-based) perturbations of gas-particle partitioning. Fig. 5 shows evaporation kinetics of OA upon continuous stripping of vapors under isothermal (25 °C) conditions simulated using volatility parameter values from multiple independent approaches. The simulation framework





used here is described elsewhere (Saha and Grieshop, 2016). The shaded region in Fig.5 shows the prediction range applying dual-TD derived parameter values from this study within estimated uncertainty ranges (campaign average and unified fits of $f_i$, $\gamma_e = 0.1$ to 1) with initial $C_{OA}$ values from 2 to 10 µg m$^{-3}$ and $d_p = 100$ nm and 150 nm. Simulations are also shown with the OA volatility distribution from Hu et al.(2016) and FIGAERO-CIMS-derived OA volatility distribution (Lopez-Hilfiker et al.,

2016) from Centreville measurements. The room temperature evaporation data from Vaden et al. (2011) measurements of ambient aerosols during the Carbonaceous Aerosols and Radiative Effects Study (CARES-2010) field campaign in Sacramento, California are also shown. This study attributed the observed slower-than-expected evaporation to extreme kinetic limitations to mass transfer ($\gamma_e \ll 0.1$). Although a direct comparison of observations collected in California and simulations based on volatility distributions from Centreville is not ideal, the consistency of volatility behavior across our and other sites

(Fig. 2; Table 1) suggests it is reasonable. Fig. 5 shows that these data fall within the range of values simulated using our TD-estimated volatility parameter values ($\gamma_e \geq 0.1$). The Hu et al. (2016) volatility distribution with $\gamma_e = 1$ also recreates these data. In contrast, simulations with the FIGAERO-CIMS-derived OA volatility distribution (Lopez-Hilfiker et al., 2016) from Centreville measurements (assuming $\gamma_e = 1$) predict almost zero evaporation (dashed black line in Fig. 5). This distribution thus appears to be inconsistent with our observations and those from room temperature evaporation experiments.

### 3.4 Temporal variation of OA volatility

A time series of OA volatility distributions extracted over the campaign period is shown in Figs. 6 (Centreville) and S10 (Raleigh). The volatility distributions ($f_i$) were extracted as described above from ~ 6-hour windows with fixed $\Delta H_{vap} = 100$ KJ mol$^{-1}$ and $\gamma_e = 0.5$ based on the best estimates from campaign-average fits. The average and (95% confidence interval) of $\overline{C^*}$ (µg m$^{-3}$) are 0.18 (0.05 - 0.54) and 0.16 (0.04 - 0.43) for the Centreville and Raleigh data sets, respectively, in line with

values from the campaign-average and unified fits. The OA volatility distributions do not vary dramatically over the campaign period for either site. Ambient OA concentrations ($C_{OA}$) shown in Fig. 6a (Centreville) and S10a (Raleigh) suggest that there was no consistent relationship between $C_{OA}$ and OA volatility. Overall, there is a no apparent trend in OA volatility for either site.

Fig. 6b shows a time series of the fractional contribution of isoprene-derived OA and more-oxidized oxygenated OA (MO-

OOA) (Xu et al., 2015a, 2015b) to total OA during the Centreville campaign. Isoprene was the dominant biogenic VOC (> 80% of total VOC mass) measured during Centreville campaign (Xu et al., 2015b), and is the biogenic VOC with greatest global emissions (Sindelarova et al., 2014). Isoprene-derived OA contributed ~ 17-18% to the campaign average $C_{OA}$ at the Centreville site during the SOAS (Hu et al., 2015; Xu et al., 2015a, 2015b) while MO-OOA contributed ~ 39% (Xu et al., 2015a, 2015b). Lopez-Hilfiker et al. (2016) reported isoprene-derived OA was more volatile than the remaining OA using

FIGAERO-CIMS measurements at the Centreville site. This result contradicts with Hu et al. (2016), who reported a lower volatility of isoprene-derived OA than the bulk OA using TD measurements at the same site. Since our derived volatility distributions are for the bulk OA, we cannot make a specific comment on the volatility of isoprene-derived OA. However, if





the volatility of isoprene-derived OA differs substantially from the remaining bulk, OA volatility might be expected to co-vary with the fractional contribution of isoprene-OA to $C_{OA}$. Fig. 6c shows extracted bulk OA volatility distributions and their mean $C^*$ over the Centreville campaign period. Fig.6d shows a scatter plot of mean $C^*$ versus the fractional contribution of isoprene-OA to $C_{OA}$; the two show no correlation. This result indicates that the effective volatility of isoprene-OA may not be

substantially different than the remaining bulk OA. If there is a difference we cannot differentiate this effects from bulk OA volatility, potentially due to the contributions of other components to bulk OA.

Diurnal trends in OA volatility distributions are shown in Figs. 7 (Centreville) and S11 (Raleigh). Results show that OA appeared relatively less volatile in the afternoon than early in the morning for both sites, consistent with previous field measurements in Mexico City (MILAGRO) and Riverside (SOAR-1) (Huffman et al., 2009). Fig. 7a shows diurnal trends of

10 OA factors derived from PMF analysis during the Centreville campaign (Xu et al., 2015a, 2015b). Less-oxidized oxygenated-OA (LO-OOA; average O:C ~ 0.63) dominated in the early morning (~40-50 %) while more-oxidized oxygenated-OA (MO-OOA; average O:C ~ 1.02) was the largest OA component in the afternoon (~50%). Xu et al.(2015a) hypothesized that oxidation of monoterpenes forms a large portion of observed LO-OOA in the Southeastern U.S. via $NO_x$ and $O_3$ ($NO_3$ radical) pathways, and that organo-nitrates contribute substantially to LO-OOA (20-30%). Laboratory chamber experiments also

suggest that nitrate-containing species make a significant contribution to SOA formed during terpene photooxidation/ozonolysis under high $NO_x$ conditions (Ng et al., 2007; Presto et al., 2005), and from reactions with nitrate radicals (Boyd et al., 2015). Lee et al. (2011) observed greater evaporation in a TD of α-pinene and β-pinene ozonolysis SOA formed under high NOx conditions than under low NOx condition. Thus the higher volatility observed in the morning can likely be linked with the prevalence of LO-OOA and possible contributions from organo-nitrates. In contrast, bulk OA was

dominated by MO-OOA in the afternoon. That OA is relatively less volatile in the afternoon is consistent with the observation that OA volatility often decreases with increased oxidation (during functionalization) (Jimenez et al., 2009). Fig. 8 shows scatter plots of $\overline{C^*}$ versus LO-OOA and MO-OOA fractions of OA during the Centreville campaign. Although the average slopes of the scatter plots show an increase (decrease) of $\overline{C^*}$ with increasing LO-OOA (MO-OOA) fraction, respectively, these correlations are not strong (correlation coefficient; r ~ 0.5). A poor correlation between $\overline{C^*}$ and OA factors is also observed in

the Raleigh data set. For example, Fig. S12 shows scatter plots of $\overline{C^*}$ versus tracer $m/z$ based HOA fraction and OOA fraction estimates (Ng et al., 2011b) with an average slope of $-0.3 \pm 0.16$ (r ~ 0.2) for HOA and $-0.12 \pm 0.11$ (r ~ 0.1) for OOA. The observed link between OA volatility and oxidation state is further discussed in sec. 3.5.

## 3.5 Average volatility and oxidation state of OA

Fig. 9 explores the link between average carbon oxidation state, $\overline{OS_c}$, calculated as 2× O:C - H:C (Kroll et al., 2011), and $\overline{C^*}$.

O:C and H:C are estimated from an empirical parameterization of OA elemental ratio from unit mass resolution data, given by Canagaratna et al. (2015) as a function of $f_{44}$ (O:C = 0.079 + 4.31× $f_{44}$ ) and $f_{43}$ (H:C = 1.12 + 6.74 × $f_{43}$ − 17.77 × $f_{43}^2$) , respectively. $f_{44}$ and $f_{43}$ are the fractional ion intensity at $m/z$ 44 and 43, respectively, taken from ACSM measurements. The



estimated OA elemental ratios using the above empirical parameterizations are in relatively good agreement with those determined via elemental analysis of the high resolution mass spectra data (HRToF-AMS) collected by other groups during SOAS. For example, our estimated campaign average O:C during Centreville campaign (0.68 ± 0.07) is within 1-2 standard deviation of that determined in Xu et al.(2015b) (~ 0.78).

The scatter plot of $\overline{OS_c}$ versus $\overline{C^*}$ (Fig. 9) shows a mild downward trend, which is suggestive of lower-volatility OA being associated with higher oxidation state. However, the correlation is not statistically robust (r < 0.3). This is consistent with the observations of Xu et al.(2016) and Paciga et al. (2015) who reported weak association between average oxidation state and volatility for OA measured in the London and Paris areas, respectively. The campaign-average $\overline{OS_c}$ during the Centreville measurements (-0.18 ± 0.15) was higher than in Raleigh (-0.42 ± 0.16) (p-value << 0.0001), whereas campaign-average $\overline{C^*}$ values were essentially identical (Centreville: 0.18 ± 0.14, Raleigh: 0.16 ± 0.12 µg m$^{-3}$; p-value > 0.1).

### 3.6 Application of measured volatility distribution to evaluate simulated OA in a CTM

Fig. 10 compares the measured and simulated OA volatility distributions at Centreville for June, 2013. The simulated OA volatility distribution in the $C^*$ bins between $10^0$ and $10^1$ µg m$^{-3}$ agrees reasonably well with observations. The model predicts a dominance of BSOA in the two bins, consistent with observations in the Centreville region. However, large discrepancies exist between the observed and simulated OA volatility distribution in the C* bins between $10^{-2}$ and $10^{-1}$ µg m$^{-3}$. The model tends to greatly underpredict the OA concentrations in this volatility range. WRF/Chem did not reproduce the observed portion of the mass of OA in the lower $C^*$ bins, from $10^{-4}$ to $10^{-1}$ µg m$^{-3}$, because the VBS SOA module in this version of WRF/Chem does not treat volatility in this range. Consistent with the measurement results from this study, a number of laboratory (Ehn et al., 2014; Jokinen et al., 2015; Kokkola et al., 2014; Zhang et al., 2015) and field (Hu et al., 2016; Lopez-Hilfiker et al., 2016) studies have reported a significant fraction of SOA from biogenic precursors is low-volatility. These low volatility materials are missing in the WRF/Chem simulation.

The simulated total OA mass concentration ($C_{OA}$) was underpredicted by a factor of 2 to 3 at Centreville during the SOAS period. Several factors may contribute to this underprediction. Comparison of WRF/Chem predictions of most relevant meteorological variables and major precursor VOCs with measurements collected during the SOAS shows a relatively good performance (Yahya et al., 2016a). For example, the mean biases for simulated temperature at 2 m, relative humidity at 2 m, and wind speed at 10 m are -0.9 °C, -0.8%, and 0.3 m s$^{-1}$, respectively. The normalized mean bias (NMB) of the simulated planetary boundary layer height (PBLH) is -38%, which would tend to bias OA concentrations high, suggesting that the underprediction in PBLH is not responsible for the underpredictions of OA. In terms of VOC concentrations, the model performs well for β-pinene and formaldehyde with NMBs of -8.5% and -4.3%, respectively, but underpredicts α-pinene with an NMB of -51.7% and significantly overpredicts limonene with an NMB of 249% (Figure not shown). The WRF/Chem simulation only considers the SOA formed from a few BVOCs including isoprene, α-pinene, β-pinene, limonene, humulene, and ociene and does not account for contributions from other BSOA precursors such as other sesquiterpenes. Therefore,





underestimation of precursor VOC emissions and missing precursors may contribute to OA underprediction. Other sources of uncertainty in the VBS treatment in WRF/Chem include the coarse spatial resolution in the model simulation, the assumed fraction of OA added for each oxidation/aging step, the assumed fragmented and functionalized percentages of organic condensable vapors, as well as the uncertainties in the dry and wet deposition velocities of SOA and SOA precursors. These

factors can also contribute to the discrepancies between the model and observed $C_{OA}$ at Centreville.

One likely contributor to the model's under-prediction is issues with the SOA yield parameterizations in the model. Smog chamber growth-experiment-derived mass yield coefficients (i.e., distributions of product mass yield in different volatility/$C^*$ bins) (Pathak et al., 2007) are used to model SOA in a CTM. The estimated SOA yield from a traditional smog chamber experiment could be underestimated due to wall losses of condensable vapors. For example, Zhang et al. (2014) showed up to

a factor of 4 yield underestimation for toluene SOA due to this fact. The high and low $NO_x$ mass yields used in WRF/Chem simulations for ASOA and BSOA are based on traditional smog chamber yield experiments, taken from Lane et al.(2008b). These distributions do not consider mass yields from the $C^*$ bins $10^{-4}$ to $10^{-1}$ μg m$^{-3}$, where a significant portion of the OA mass was observed. The substantial amounts of low volatility materials are typically missing in these traditional yield measurement based distributions (Kolesar et al., 2015b; Saha and Grieshop, 2016). Our recent dual-TD-based effort to

determine the SOA mass yield distribution for α-pinene ozonolysis (Saha and Grieshop, 2016) indicates products are substantially less volatile than the parameterizations used in current models (including that discussed above). This α-pinene product distribution suggests a factor of 2-4 more SOA yield under atmospherically relevant conditions compared to traditional distributions from smog chamber growth experiments. Updating SOA mass yield coefficient data is likely required for all known precursors, and may lead to large improvements in model predictions of both $C_{OA}$ and OA volatility distributions.

The WRF/Chem simulation used the semi-empirical $\Delta H_{vap}$ correlation derived by Epstein et al. (2010) ($\Delta H_{vap} = 130 - 11\log_{10}C^*_{, 298K}$), which gives higher values, with a steeper $\log_{10}C^*$ dependence, than TD-derived values. Since WRF/Chem-predicted $C_{OA}$ was a factor of 2-3 underestimated with the Epstein et al. (2010) $\Delta H_{vap}$ values, use of lower values (e.g., 100 kJ mol$^{-1}$ as suggested by our TD observations) will further increase the discrepancy between observed and simulated OA concentrations for a given mass yield distribution.

**4. Conclusions and Implications**

This paper presents results from ambient OA volatility measurements from two sites in the southeastern U.S. under diverse conditions. Measurement campaigns were conducted at a BVOC-dominated forested rural setting during summer and another more anthropogenically-influenced, but forested urban location under cooler conditions. This study applied a dual-thermodenuder (dual-TD) setup that varied temperature and residence time in parallel. Ambient OA gas-particle partitioning

parameters ($C^*$, $\Delta H_{vap}$, $\gamma_e$) value were extracted by fitting observed dual-TD data using an evaporation kinetic model. The OA volatility distribution derived via inverse modeling is sensitive to $\Delta H_{vap}$, and $\gamma_e$ values. The addition of variable residence time



TD (VRT-TD) data provided tighter constraints on the extracted parameter values. A $\Delta H_{vap}$ of ~ 100 KJ mole$^{-1}$ and $\gamma_e$ of 0.5 best explain observations collected at both sites, under diverse conditions. An effective $\gamma_e$ value of ~ 0.1 to 1 can explain observed evaporations within variability while a very small $\gamma_e$ value ($\gamma_e \ll 0.1$) cannot fit the observations from both TDs. The Epstein et al. (2010) $\Delta H_{vap}$ correlation, which was determined based on measured properties of a variety of known compounds

also did not reproduce the evaporation observed in this study.

While measurement campaigns were conducted under different meteorological conditions at locations with differing levels of biogenic and anthropogenic emissions, the derived OA volatility distributions are found to be very similar. A substantial amount of OA (40-70%) at both sites was found to be of very low volatility ($C^* \leq 0.1$ µg m$^{-3}$) so will remain predominantly in the particle-phase (effectively non-volatile) under typical atmospheric conditions. OA volatility distributions also did not vary

substantially over the campaign period. Our derived OA volatility parameterizations appear to be broadly consistent with observations of room temperature evaporation (Vaden et al., 2011) during CARES-2010 in California. The observed consistency in OA volatility across diverse settings is an important finding, which implies that OA in the atmosphere formed from a variety of sources can exhibit similar volatility properties and chemical signatures. This result also suggests that measurements of OA volatility distributions such as derived here could provide good diagnostics for overall model

representativeness, but may not be as useful for diagnosing differences across sites and conditions.

The diurnal profile of extracted OA volatility showed that bulk OA was relatively less volatile in the afternoon than early in the morning. This trend is consistent with the prevalence of LO-OOA (less oxidized) in the morning and MO-OOA (more oxidized) in the afternoon. However, while average O:C and/or oxidation state ($\overline{OS_c}$) of bulk OA is often considered linked to volatility, in our data sets correlations between mean oxidation state ($\overline{OS_c}$) and mean volatility ($\overline{C^*}$) were weak (r <0.3). This

observed weak correlation and the fact that atmospheric OA is a complex mixture of organics of a broad range of volatilities and oxidation states, reinforces the need to measure and understand the distribution of both volatility and oxidation states. The 2D-VBS framework (Donahue et al., 2012) offers one way to constrain these parameters in atmospheric models. While determination of OA volatility distributions was the focus of this study, future efforts also should measure distributions of volatility and oxidation states comprising ambient OA.

The gas-particle partitioning parameters ($C^*$, $\Delta H_{vap}$, $\gamma_e$) extracted from these measurements have important implications for the treatment and evaluation of OA in current atmospheric models. Since a CTM incorporating the VBS framework predicts OA concentrations in each volatility ($\log_{10}C^*$) bin (i.e., OA volatility distribution), comparison of simulated and measured OA volatility distribution is an useful means for model evaluation beyond only comparing total OA concentration ($C_{OA}$). Here, we compared our measured OA volatility distribution with that simulated by WRF/Chem. To our knowledge, this is the first direct

evaluation (with observations) of OA volatility distributions simulated in a CTM using the VBS framework. This evaluation indicates that OA volatility distributions predicted in WRF/Chem are inconsistent with measurements over the $C^*$ range from $10^{-4}$ to $10^{-1}$ µg m$^{-3}$. This may give important clues towards the root causes of the model's underestimation of $C_{OA}$ by a factor of 2 to 3. In comparison to our TD-derived OA volatility distribution and other recent evidence (Ehn et al., 2014; Hu et al.,





2016, 2016; Jokinen et al., 2015; Kokkola et al., 2014; Lopez-Hilfiker et al., 2016; Saha and Grieshop, 2016), low-volatility materials are mostly missing from the WRF/Chem predictions. Recent evidence of SOA from aqueous-phase oxidation in presence of abundant particle water (Carlton and Turpin, 2013; Marais et al., 2016), formation of oligomers and large molecular compounds directly in the gas- phase (Ehn et al., 2014) and via condensed phase chemistry (Kroll et al., 2015; Kroll and Seinfeld, 2008) suggest that complex and multi-phase formation and evolution processes produce SOA in the atmosphere. Many of these processes can produce very low-volatility organics and most are not included in current CTMs. These low-volatility organics appear to make significant contributions to the atmospheric OA budget and cloud condensation nuclei formation (Jokinen et al., 2015).

The $\Delta H_{vap}$ and $\gamma_e$ values extracted here for atmospheric OA in the Southeastern U.S. also have important implications for predicting OA concentrations in a CTM. First, a $\Delta H_{vap}$ value of 30-40 KJ mol$^{-1}$ (Farina et al., 2010; Lane et al., 2008b; Pye and Seinfeld, 2010) is typically assumed for modeling OA in a CTM, which is substantially lower than that suggested by our TD observations (~100 KJ mol$^{-1}$ ). An increase of assumed $\Delta H_{vap}$ value will increase atmospheric OA burden and lifetime for a particular input volatility distribution (Farina et al., 2010). Finally, a value of $\gamma_e \geq 0.1$ indicates a gas-particle repartitioning timescale (Saleh et al., 2013) on the order of minutes to an hour under atmospherically relevant conditions ($N_p \sim$ 1000-5000 cm$^{-3}$). Therefore, the equilibrium phase-partitioning assumption typically made in CTMs should be reasonable for a prediction timestep of ~ 1 hour.

## Acknowledgements

We thank Satoshi Takahama and his research group at EPFL for their help and support during the SOAS campaign, Paul Shepson's group (Purdue University) for BVOC data, SEARCH Network for temperature, relative humidity, mixing height, CO, NO$_x$ data from the Centreville site. Operation of the SEARCH network and analysis of its data collection are sponsored by the Southern Co. and Electric Power Research Institute.

Funding was provided by start-up support from North Carolina State University, Raleigh, USA. AK acknowledges funding by USEPA (grant 83541101). Contents of this publication are solely the responsibility of the authors and do not necessarily represent the official views of the USEPA. Further, USEPA does not endorse the purchase of any commercial products or services mentioned in the publication. KY and YZ acknowledge funding from the National Science Foundation EaSM program (AGS-1049200) for WRF/Chem simulations and high-performance computing support from Stampede, provided as an Extreme Science and Engineering Discovery Environment (XSEDE) digital service by the Texas Advanced Computing Center (TACC) (http://www.tacc.utexas.edu), which is supported by National Science Foundation grant number ACI-1053575 and Yellowstone (ark:/85065/d7wd3xhc) provided by NCAR's Computational and Information Systems Laboratory, sponsored by the National Science Foundation and Information Systems Laboratory. LX and NLN acknowledge National Science Foundation grant 1242258 and US Environmental Protection Agency STAR grant RD-83540301.



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

**Table 1:** Best fit OA volatility parameter values extracted from this study along with several previous field and lab studies

| Study | Centreville (this study) | | Raleigh (this study) | | FAME (Lee et al., 2010) | | MILAGRO (Cappa and Jimenez, 2010) | | | | AP-SOA (Saha and Grieshop, 2016) |
|---|---|---|---|---|---|---|---|---|---|---|---|
| Campaign average $C_{OA}$ (µg m$^{-3}$) | 5.2 | | 6.7 | | 2.8 | | 17 | | | | 5 |
| Note | a | b | a | b | | | c | d | c | d | e |
| $\gamma_e$ | 0.5 | 0.5 | 0.5 | 0.5 | 0.05 | 1 | 1 | 1 | 0.1 | 0.1 | 0.1 |
| $\Delta H_{vap}$ (KJ mol$^{-1}$) | 100 | 100 | 100 | 100 | 80 | 80 | 100 | 100 | 100 | 100 | [80,11][‡] |
| $\log C^*$ (µg m$^{-3}$) | | | | | | $f_i$ | | | | | |
| -6 | | | | | | | 0.06 | | 0.04 | | |
| -5 | | | | | | | 0.06 | | 0.04 | | |
| -4 | 0.14 | 0.18 | 0.14 | 0.16 | | | 0.06 | 0.27 | 0.04 | 0.21 | 0.03 |
| -3 | 0.05 | 0.05 | 0.06 | 0.05 | | 0.2 | 0.07 | 0.11 | 0.04 | 0.07 | 0.07 |
| -2 | 0.06 | 0.08 | 0.08 | 0.13 | 0.2 | 0.2 | 0.07 | 0.11 | 0.05 | 0.09 | 0.03 |
| -1 | 0.15 | 0.13 | 0.12 | 0.20 | 0.2 | 0.3 | 0.08 | 0.12 | 0.06 | 0.10 | 0.12 |
| 0 | 0.29 | 0.33 | 0.28 | 0.20 | 0.3 | 0.3 | 0.10 | 0.15 | 0.1 | 0.18 | 0.18 |
| 1 | 0.31 | 0.23 | 0.32 | 0.26 | 0.3 | | 0.16 | 0.24 | 0.2 | 0.35 | 0.57 |
| 2 | | | | | | | 0.34 | | 0.43 | | |
| Mean $C^*$ (µg m$^{-3}$) | 0.21 | 0.12 | 0.20 | 0.10 | 0.50 | 0.05 | 0.32 | 0.03 | 1.5 | 0.1 | 1.16 |
| $C_{sat\_eff}$ (µg m$^{-3}$) | 1.8 | 1.4 | 2.0 | 1.5 | 1.3 | 0.3 | 9.4 | 1.9 | 13.8 | 2.8 | 3.5 |

5   [a] Campaign average dual-TD data fit with campaign average $C_{OA}$ and $d_p$.

[b] Unified fit of individual measurement from whole campaign (*MFR*, $C_{OA}$, *dp*; 20-30 minute resolution data).

[c] The $f_i$ distribution derived from $C_{i,tot} = a_1 + a_2 exp[a_3(log(C^*)-3)]$; $f_i = C_{i,tot} /\Sigma C_{i,tot}$ ; $a_1$, $a_2$, and $a_3$ coefficients were taken from table 1 of Cappa and Jimenez (2010). $\log_{10}C^*$ bin ranged from -6 to +2, as in Cappa and Jimenez (2010).

[d] Same as c, but only considered $\log_{10}C^*$ bin range of -4 to +1 to be consistent with the bin ranges used in this study. To do so,

10   materials in $\log_{10}C^* < -4$ bins are assigned to -4 bin, material at $\log_{10}C^* = 2$ bin is excluded, and distribution is renormalized to make $\Sigma f_i = 1$.

[e] Chamber generated SOA from low-$C_{OA}$ α-pinene ozonolysis experiment applying renormalization approach described in note e to the distribution given in Saha and Grieshop (2016) SI, table S.5.

[‡] $\Delta H_{vap}$ (KJ mol$^{-1}$) = 80-11 $logC^*$ (µg m$^{-3}$)





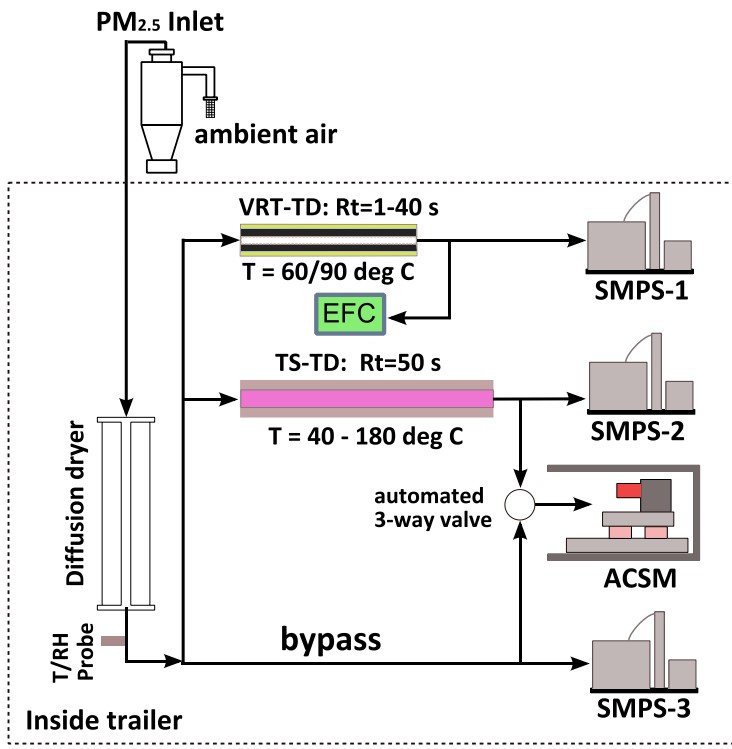

**Figure 1:** Dual thermodenuder aerosol volatility measurement setup used during field campaigns at two sites in the southeastern U.S. TS-TD: Temperature stepping TD, VRT-TD: Variable residence time TD, Rt: Residence time, EFC: Extra flow control, ACSM: Aerosol chemical speciation monitor, SMPS: Scanning mobility particle sizer.



**Figure 2:** Measured (solid symbols) and modeled (solid thick lines) campaign average organic aerosol (OA) mass fraction remaining (MFR) as a function of TD temperatures (T) and residence times (Rt). The solid symbol shows mean value and error bar is ± one standard deviation of all campaign data at each (T, Rt) condition. Raw data at each (T, Rt) condition are averaged over 20-30 minutes. Model lines are shown using the 'best fit' volatility parameter values from campaign average TD data fit (parameter values listed in Table 1). TD measurement data from the Centreville site collected by the University of Colorado group at SOAS-2013 (Hu et al., 2016) are also shown. Measurements from several previous field studies are shown with various open symbols: Hyytiala/2008-2010, Finland (Häkkinen et al., 2012); ClearfLo/2012, London (Xu et al., 2016); MILAGRO/2006, Mexico City (Huffman et al., 2009); SOAR-1/2005, Riverside, California (Huffman et al., 2009); FAME/2008, Finokalia, Greece (Lee et al., 2010); MEGAPOLI/2009-10, Paris, France (Paciga et al., 2015). Chamber alpha-pinene SOA (dark ozonolysis, COA ~ 5 μg m-3, VMD ~ 140 nm) evaporation data are shown from Saha and Grieshop (2016). In panel-a, data are color coded by TD residence times used during measurements.



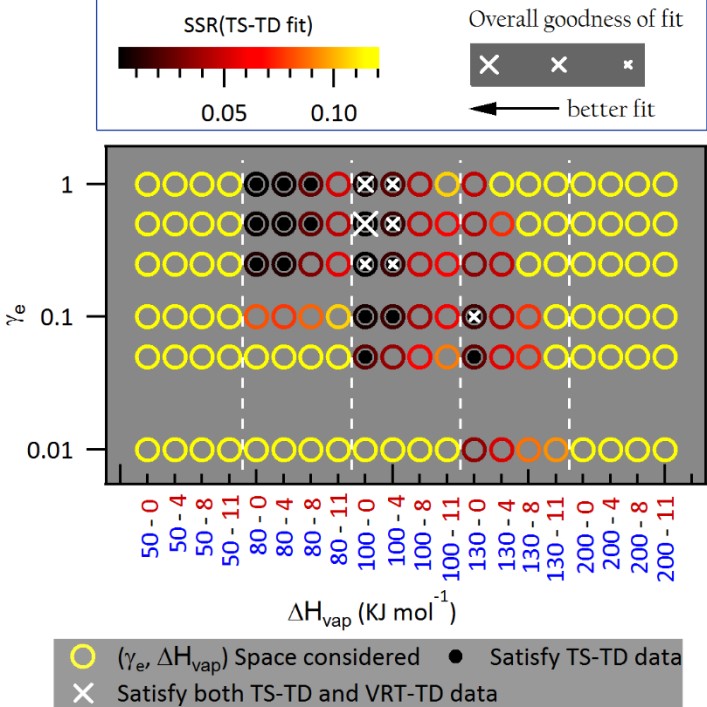

**Figure 3:** Extraction of OA gas-particle partitioning parameter ($\Delta H_{vap}$, $\gamma_e$ and $f_i$) values via evaporation kinetic model fits to campaign-average dual-TD observations during the Centreville campaign. A relationship of $\Delta H_{vap}$ = intercept-slope ($\log_{10} C^*$) was assumed (e.g., 50-0 on x-axis represents intercept =50 and slope = 0). Symbols and colors represent the goodness of fit. Points with filled inner circles recreate TS-TD observations and points with a white cross (x) recreate both TD data sets to within observational variability. Crosses represent the overall goodness of fit including both TS-TD and VRT-TD observations, with larger size corresponding to a better fit.





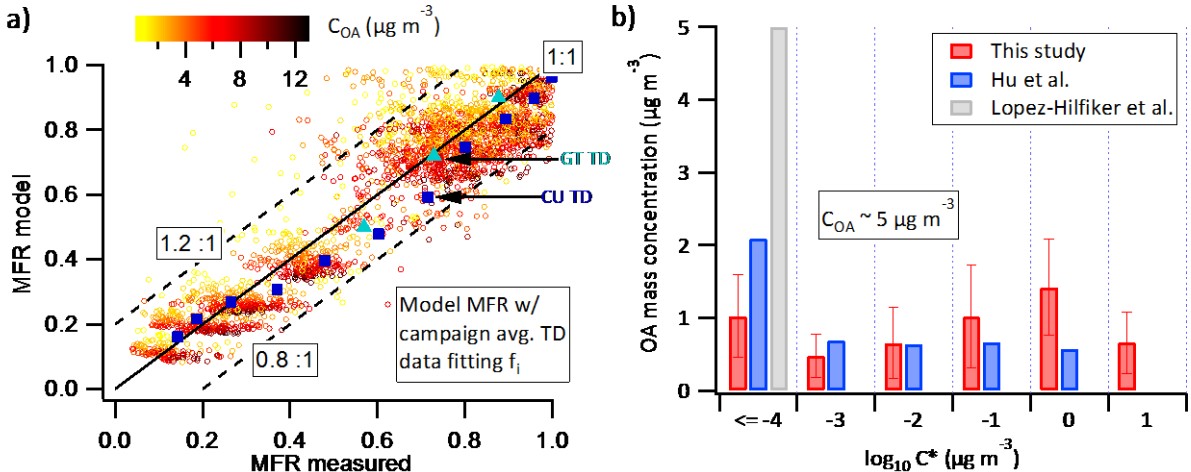

**Figure 4:** (a) Comparison of individual observations from the Centreville campaign and corresponding modeled MFRs applying the extracted $f_i$ distribution from the campaign-average fit. MFR data collected by other groups during the Centreville campaign are also shown: University of Colorado TD (CU TD; blue squares)(Hu et al., 2016) and Georgia-Tech TD (GT TD; cyan triangles) (Cerully et al., 2015) along with corresponding MFRs modeled applying volatility parameterizations from this study with the campaign average $C_{OA}$ and $d_p$. Fig. S9 shows an extended data figure of panel a, including similar plot using the $f_i$ distribution from the unified fit and analysis results for the Raleigh data set. (b) Comparison of the SOAS campaign-average OA volatility distribution (showing only condensed phase) derived from this study (dual-TDs; kinetic evaporation model fits), Hu et al.(2016) (TD; method of Faulhaber et al.(2009)), and Lopez-Hilfiker et al.(2016) (FIGAERO-CIMS). Error bars on data from this study are ± one standard deviation of distributions extracted over the campaign period (Fig.6).





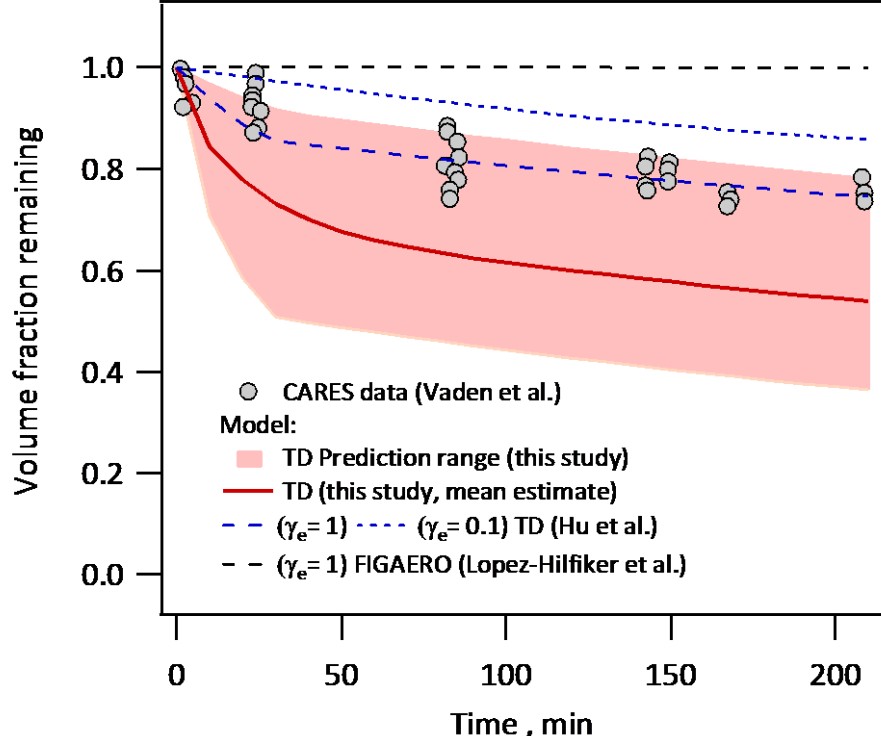

**Figure 5:** Isothermal evaporation kinetics of OA at 25˚C (room temperature) upon continuous stripping of vapors. Shaded region shows the evaporation kinetic model prediction range applying TD-derived volatility parameter values from this study; solid line shows the mean estimate. Dashed lines show model predictions using the OA volatility distribution derived using alternative approaches during the Centreville campaign (Hu et al., 2016; Lopez-Hilfiker et al., 2016). Symbols show experimental data from Vaden et al. (2011) collected during the CARES-2010 field campaign in California.





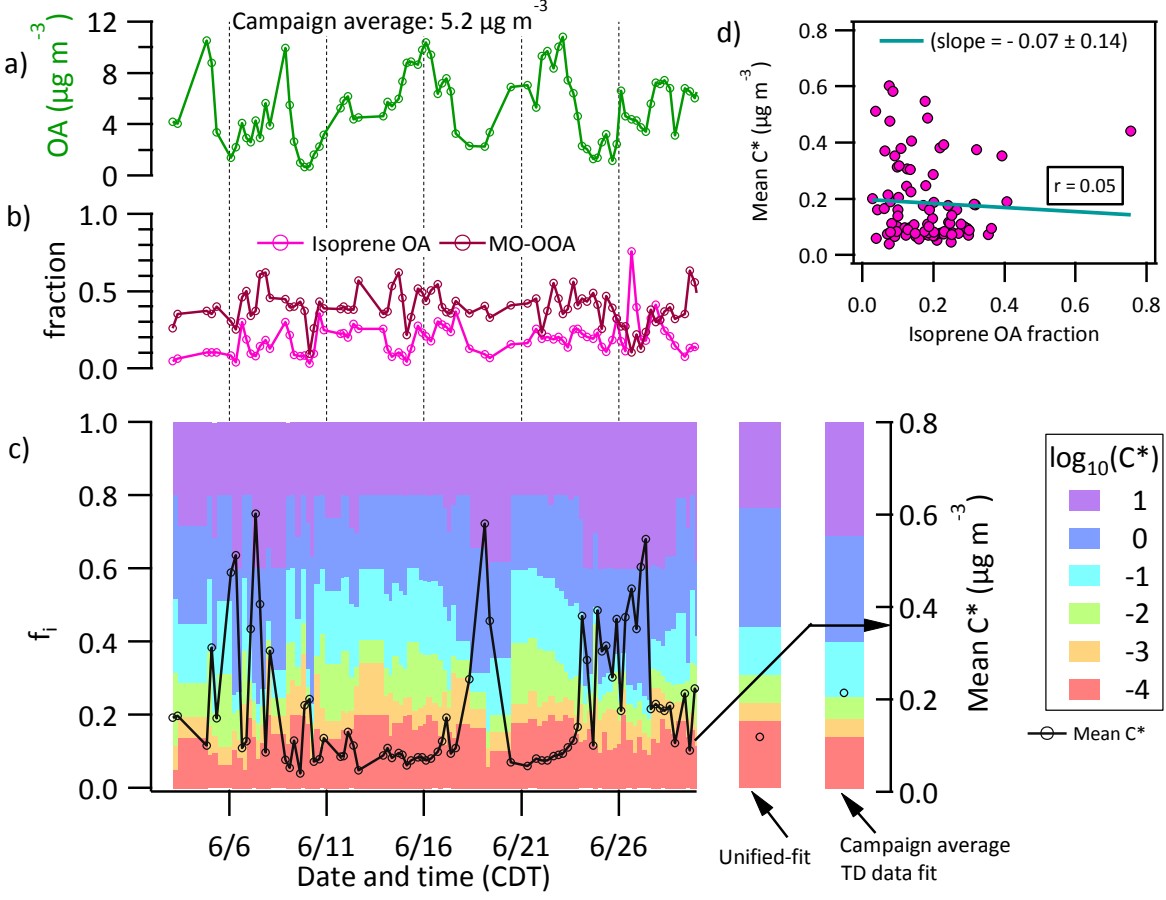

**Figure 6:** Time series of (a) ambient organic aerosol concentrations; $C_{OA}$, (b) fractional contribution of isoprene OA and more-oxidized oxygenated OA (MO-OOA) to total OA determined from PMF analysis, and (c) OA volatility distribution ($f_i$) and $\overline{C^*}$ (open black circles) during the Centreville campaign. All data are averaged over ~ 6 hours (the time resolution of $f_i$ distribution). Panel (d) shows a scatter plot of $\overline{C^*}$ verses isoprene-OA fraction in $C_{OA}$. Fig. S10 shows similar analysis results for the Raleigh data set.

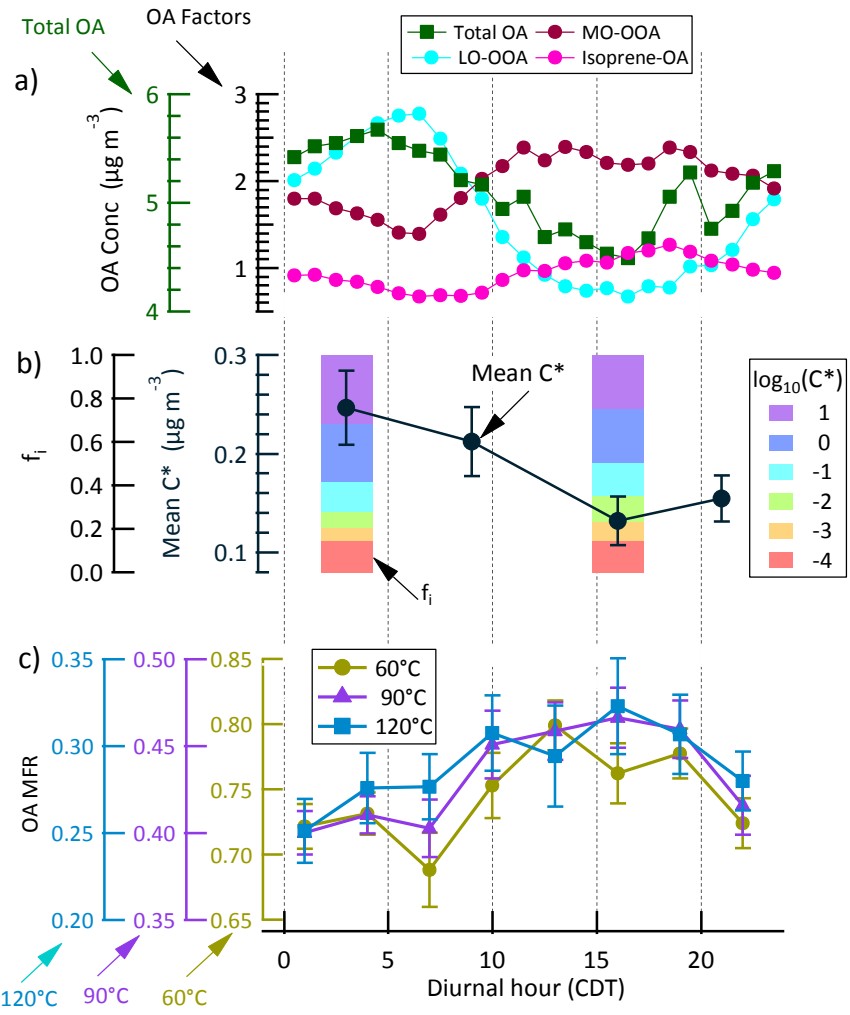

**Figure 7:** Campaign average diurnal trends for the Centreville measurements of: (a) concentrations of total OA and OA factors, (b) OA volatility ($f_i$ and $\overline{C}^*$), (c) OA MFR after heating at 60, 90 and 120 °C with a TD residence time of 50 s. Fig. S11 shows similar analysis results for the Raleigh data set. PMF factors in panel-a are LO-OOA: less-oxidized oxygenated OA; MO-OOA: more-oxidized oxygenated OA; Isoprene-OA: isoprene-derived OA (for details on OA factors analysis see Xu et al., 2015a, 2015b).





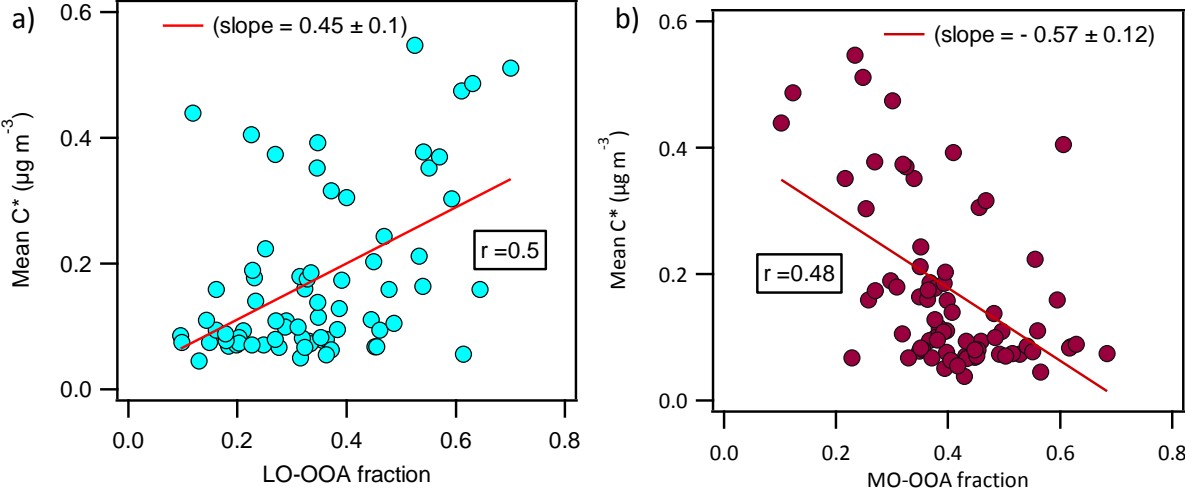

**Figure 8:** Scatter plot of mean C* verses (a) LO-OOA fraction, and (b) MO-OOA fraction in total OA concentration during the Centreville campaign.





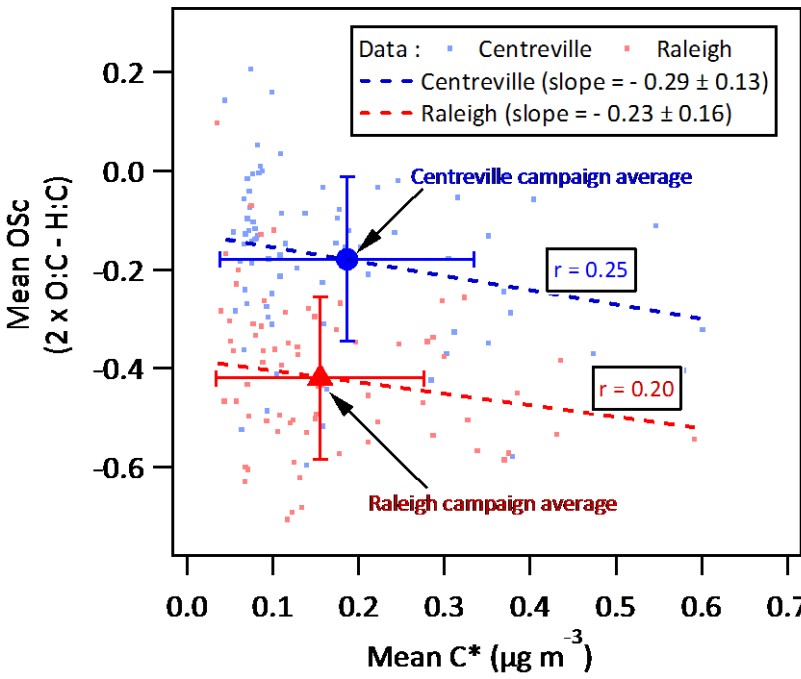

**Figure 9:** Mean oxidation state $(\overline{OS_c})$ versus mean volatility $(\overline{C^*})$ measured during the Centreville and Raleigh campaigns. Dots are campaign data, dashed lines are linear regression fits of data, and symbols are the campaign average with error bar showing ±one standard deviation.





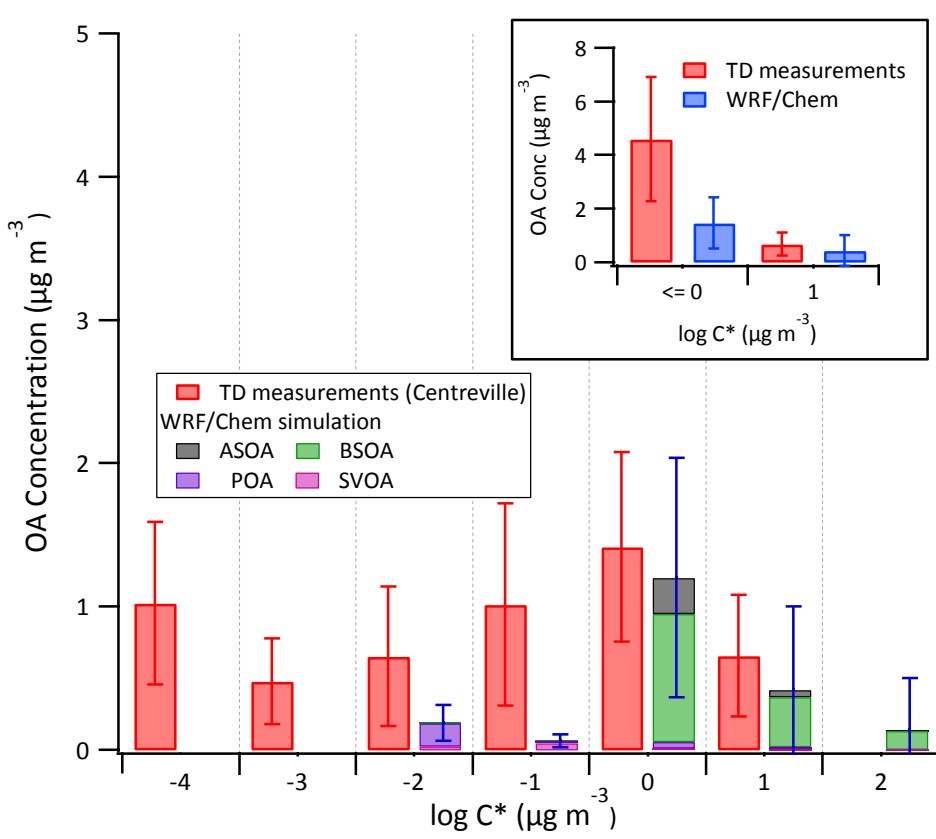

**Figure 10:** Comparison between measured OA volatility distributions and those simulated in WRF/Chem over the Centerville region. Bar height is mean, and error bar is ± one standard deviation of distributions extracted from measurements and simulations for June 2013. The inset shows a two-bin comparison (bin-1: $C^* \leq 1$ µg m$^{-3}$ and bin-2: $C^* = 10$ µg m$^{-3}$). Simulated OA components include ASOA (anthropogenic-SOA), BSOA (biogenic-SOA), POA (primary-OA), and SVOA (semi-volatile OA/oxidized POA).