# Peer review of "Quantifying the volatility of organic aerosol in the southeastern U.S."

_Atmospheric Chemistry and Physics, 2016_

## Referee Comment (RC1) · Anonymous Referee #1 · 9 Aug 2016

General Comments:

Saha and co-authors present a well-articulated investigation of the volatility properties of organic aerosol at two sites in the southeast United States. The opportunity at Centerville, AL during SOAS is uniquely favorable since there were so many collocated measurements (of meteorological metrics and AMS factors, for example). The authors do a good job motivating the juxtaposition of these data to other data they have collected in Raleigh, albeit at a different time of year. Overall, I find the goal of the work and its presentation quality to be acceptable for ACP and important for the scientific community. However, I have some concerns about the discussion. For example, I think the issue of solubility, while not specifically addressed by their measurements, should be better woven throughout their discussions in the manuscript to make it clearer that the properties of volatility and solubility simultaneously affect the gas-particle partition-

ing of organic compounds. I look forward to discussing the following issues with the authors:

Specific Comments:

1. Page 2, Line 15: Although it is well-known that volatility plays a pivotal in gas-particle partitioning, I think it is worth mentioning that solubility will also be critical for this phenomenon, especially in places like the southeast US. Admittedly, solubility is outside the scope of this paper (and the RH of the observations is maintained relatively low at 30-40%). It is of course still vital and useful to gain an in-depth knowledge of the partitioning behavior of compounds to the "dry" organic phase. However, please also be careful throughout the text about statements like in line 17-18: "vapor pressure determines whether an organic compound is found in the particle- or gas-phase" since this is not exactly true for many compounds, which are highly volatile and highly soluble, for example. This should be reworded for completeness.

2. Did the authors size select the particles before entry to the TD units? It does not appear so. Could they comment on their method of incorporating the size distribution information into their model? Did they use a moving sectional, fixed sectional, or a modal algorithm? Or was only a single diameter used as is implied on page 6, line 13? If this is the case, can the authors provide some insight into how the inaccuracies introduced in their model results from the width of the distributions (as seen in Fig. S6c,d)? Are there significant size-dependent particle losses in the system and are these temperature dependent?

3. Page 6, Line 19-21: what is the reason for putting only the fi sets that have been "accepted" against the TS-TD data through the VRT-TD analysis? What sets are unde-tected by not doing both applications for every set and then taking the best performers? On a related note, what initial fi values were used as input to the solver and how de-pendent was the "accepted" solution on these values? Were these initial conditions varied at all? Why are the authors confident they have evaluated the entire space of

volatility distributions?

4. Page 7, Line 8-9: How do the enthalpies of vaporization used in this WRF-Chem simulation compare with those derived from the observations here (could they add this to the methods section)? What are the implications of comparing the volatility distribution from a model to observations when they are using different enthalpies? Is this something that other researchers should take seriously when comparing model output to TD data? On Page 15, Lines 20-24, the authors provide some discussion of this issue. However, they seem to be assessing the sensitivity of the enthalpies in the model independently of the mass yields being used. Is this appropriate?

5. Page 9, Line 8-9: I'm not familiar with the term "condensation sink diameter". Could the authors please explain it and potentially provide an equation in the supporting info for this quantity? How is it related to the more common term, condensation sink which is in units of inverse time?

6. Do the authors have an explanation for why the 80-0 and 80-4 cases with evaporation coefficient equal to 0.5 and 1.0 performed acceptably for the Raleigh cases and not for the Centerville cases? How close were they to being accepted with the VRT-TD data? It perhaps appears that they performed better for the TS-TD data at Centerville than they did at Raleigh (or at least the cases around them with discernible colors did).

7. Page 10, Lines 5-10: The low enthalpies used in models are based on observation data (e.g. Offenberg et al. 2006; Pathak et al. 2007; Stanier et al., 2008) of so-called "effective" enthalpies, so please consider mentioning this for completeness; Stanier et al. (2008) were a bit higher than others and in line with the lower bound explored in this study. It has been argued that low enthalpies of vaporization result from describing systems with too few volatility surrogates (Riipinen et al., 2010). Could the authors comment on what they attribute the void between their high enthalpies and historical low enthalpies to? If they run their model with only two surrogates, for example, do they get low enthalpies as well?

8. Can the authors please provide some statistics to go along with the comparison in Fig. 4? For example, it would be useful to have mean bias, correlation coefficient and root mean square error so that future studies would have something succinct to use as a benchmark.

9. In Fig. 4a, is the COA that of the ambient (unheated) sample? There does appear to be systematic overprediction for low COA cases compared to high COA cases. Is this true? Could the authors provide some statistics stratified by COA to assess this? If there is a relationship, could they comment on why it emerges from their approach?

10. Page 13, Line 8: Please avoid using the word "relatively" and opt instead for a quantification of the difference between the afternoon and early morning.

11. Sections 3.4-3.5: The authors have defined both mean(C*) and C*eff and chose the former for their analysis of time series and diurnal profiles. I think this may have been an unfortunate choice. This value, as they say, accounts for both particle- and gas-phase compounds. It is not very surprising to me that it would be a more stable quantity, throughout the SOAS campaign at least. I wonder if they would capture the trends they are after better by using the C*eff which just accounts for the particle-phase material. This at least would seem to make more sense for correlating with AMS factors (Isoprene-OA, LO-OOA, and MO-OOA) since those apply just to the particle phase and not the total organic burden. Using this alternative approach would theoretically introduce a temperature and dilution dependence that might result in a more striking and meaningful variability. Of course, it may on the other hand yield a rather invariant trend similar to what they have already shown.

Minor Changes/Typos:

Pg 1, Line 11: Consider mentioning solubility as well. "quantitative estimates of the thermodynamic (volatility, water solubility, etc)"

Pg 1, Line 14: "one at a biogenic..."

Pg 2, Line 6-7: Of course SOA has also been shown to be introduced from oxidation of primary SVOCs and IVOCs followed by condensation and aqueous phase reactions of high soluble compounds. Depending on your definition of "secondary", it is also formed by heterogeneous oxidation of POA without evaporation. Please mention these here or don't be so specific about the VOCs role.

Pg 5, Line 16-17: The VBS approach does not assume unity activity coefficients. Instead the activity coefficient are assumed to be lumped into the Cvap, making it C*. And so the activity coefficients have a value, but that value is assumed not to change from the lab or field to other mixtures or conditions.

Page 6, Line 10: I disagree that the model is applied in an "inverse sense", as it appears that the authors are solving the evaporation problem in the forward way, brute-force, for all of their free parameter combinations and then comparing output to the observations. This could perhaps be labelled reverse engineering, but I think associating it with inverse modeling is inaccurate.

Page 6, Line 27: "common means to improve the performance of OA prediction in chemical transport models"

Fig. 2b,c: There is no explanation or legend for the colors of the trends. The text claims they distinguish different residence times but isn't this described by the x-axis? Please clarify this point.

Page 9, Line 14: Just write "volatility". Having the "/C*" is unnecessary. Page 15, Line 7 as well.

Page 9 and Fig. 3: Neither the description in the text nor the figure caption should claim that the figure depicts the fi values. As far as I can tell, it does not.

Page 9, Line 18: Please indicate here, and throughout the text, that this C* is at Tref (presumably 298 or 300 K). This should especially be made clear for any references to mean C*.

Page 10 and Fig. 2: The notation for Ceff is not consistent with Csat_eff.

Page 13, Line 26-27: You do not really need this final sentence since the section you refer to comes directly next.

Page 16, Line 16: Please be quantitative rather than saying "Relatively less volatile".

Page 16, Line 30: Murphy et al. (2011) compared predictions with the 2D-VBS in a Lagrangian column CTM against the FAME data reported by Lee et al.

References Murphy, B. N., Donahue, N. M., Fountoukis, C., and Pandis, S. N. (2011): Simulating the oxygen content of ambient organic aerosol with the 2D volatility basis set, Atmos. Chem. Phys., 11, 7859-7873, doi:10.5194/acp-11-7859-2011.

Offenberg, J. H., T. E. Kleindienst, M. Jaoui, M. Lewandowski, and E. O. Edney (2006), Thermal properties of secondary organic aerosols, Geophys. Res. Lett., 33, L03816, doi:10.1029/2005GL024623.

Riipinen, I, Pierce, J, Donahue, N, and Pandis, S (2010): Equilibration time scales of organic aerosol inside thermodenuders: Evaporation kinetics versus thermodynamics. Atmos. Environ., 44, 5, 597-607, doi:10.1016/j.atmosenv.2009.11.022

Stanier, Donahue, and Pandis (2008), Parameterization of secondary organic aerosol mass fractions from smog chamber data, Atmos. Environ. 42, 10, 2276-2299, doi:10.1016/j.atmosenv.2007.12.042.

---

## Referee Comment (RC2) · Anonymous Referee #2 · 1 Sep 2016

General Comments:

This manuscripts presents measurements and analysis of the evaporation behavior of organic aerosol (OA) measured at two different locations – one more rural, one more urban, but both influenced by significant concentrations of BVOCs. Measurements were taken with a dual thermodenuder system in which both the temperature and residence time were varied. Main conclusions of the work include that the OA evaporation behavior was fairly similar at the two sites and did not vary much over the course of the measurement campaign, and also that much of the OA is in low-volatility bins which are currently not represented in (most) air-quality models.

The manuscript is well written and generally well argued, and overall the topic and quality of the manuscript makes it suitable for publication in ACP. However, I have a few

concerns about the analysis and interpretation as explained below which I suggest the authors should address before publication.

Major comments:

1. The authors focus on the "best fit solution" and address to some extent the sensitivity of the error (SSR) to changes in dHvap and gamma (e.g. Fig. 3). I request that they also address the sensitivity of the error to changes in the fi. I am particularly worried that the fi in the lower C* bins may not be well constrained by the data.

2. Regarding higher C* bins, on page 6 lines 1-2 the stated reason for not including C*>10 $\mu$g/m3 is that less than 5% of the materials would be present in the condensed phase at the average COA of 5 $\mu$g/m3. Common VBS bins used range from 0.1 to 100 $\mu$g/m3 for ambient OA concentration of ~10 $\mu$g/m3. Elevated OA episodes (OA > 10 $\mu$g/m3) were observed, especially at the Raleigh site (Figure S7). It would be reasonable to include the 100 $\mu$g/m3 bin in the model to account for these episodes. Have the authors investigated the effects of including higher volatility bins on the fitted model parameters? As referee #1 pointed out and according to Riipinen et al. (2010), a two-surrogate product model could lead to very different conclusions about dHvap. How does the inclusion of more and lower volatility bins affect modeled dHvap? In other words, does bin selection introduce bias in to model results (and if so, how much)?

3. Page 6 Line 21-23: Have the authors attempted to apply the two models in a different order (VRT-TD first then TS-TD)? How much does this change the results?

4. Page 13 Line 1-6: mean C* is perhaps not a suitable metric to correlate with the fraction of isoprene OA. If isoprene OA was in fact much less volatile, its overall contribution to mean C* would be minor; the calculated mean C* value would be dominated by fi values of more volatile surrogate compounds. Using Ceff would present similar issues. It seems more appropriate to investigate the correlation between isoprene OA and individual bin fi's.

5. Page 6, line 7: Was a constant collection efficiency of 0.5 also applied to all thermally denuded data? Evaporating part of the organic aerosol (and therefore changing the org/sulfate ratio) could change the collection efficiency, which would bias MFR measurements. The authors could partially address this issue by comparing total SMPS and ACSM measurements (mass), i.e. the ACMS/SMPS ratio in the bypass and after the thermodenuder.

6. Fig S7 seems to assume that all nitrate measured in the ACSM is inorganic. This seems to contradict discussion earlier in the manuscript of the potential importance of organic nitrates. Previous work has shown that the ratio of NO+ to NO2+ fragments measured by AMS or ACSM instruments is quite different for organic nitrates and ammonium nitrate, and that the ratio can be used to estimate the fraction of measured nitrate due to organic nitrates. I suggest the authors use these measurements from the ACSM to estimate how much of the measured nitrate is organic vs. inorganic. Assuming full neutralization of sulfate by ammonium (which is reasonable in the presence of ammonium nitrate) could also be used to calculate the nitrate attributable to ammonium nitrate and, therefore, the nitrate due to organic nitrates.

7. Page 4 Line 12: "All Rts reported here are calculated assuming plug flow at room temperature" This implies that a plug flow (as opposed to parabolic) velocity profile is assumed in the evaporation model. Is this correct? It was suggested by (Cappa, 2010) that assuming plug flow profile would lead to underestimation of dHavp and overestimation of Csat. Have the authors explored different assumptions for gas velocity profile in the model?

Minor comments:

8. Page 5 Line 2: Please expand on what "instrumental inter-calibration factors" entail

9. Page 6 Line 23-24: Please clarify on how variability is calculated. Is it based on measured MFR values or OA measurements? Is it calculated over small intervals or over the entire course of campaign?

[Printer-friendly version]{.underline}

[Discussion paper]{.underline}

10. Page 8 Line 31: Please provide references for previous field studies.

11. Page 10 Line 8-10: Please add discussion on why observed slope may be different from the empirical relation determined by Epstein et al.

12. Page 2 line 6: "...SOA is formed in the atmosphere via condensation of low-volatility products..." This statement is not inclusive enough. Please revise.

References

Cappa, C. D.: A model of aerosol evaporation kinetics in a thermodenuder, Atmos. Meas. Tech., 3(3), 579–592, doi:10.5194/amt-3-579-2010, 2010.

Riipinen, I., Pierce, J. R., Donahue, N. M. and Pandis, S. N.: Equilibration time scales of organic aerosol inside thermodenuders: Evaporation kinetics versus thermodynamics, Atmos. Environ., 44, 597–607, doi:10.1016/j.atmosenv.2009.11.022, 2010.

---

## Author Comment (AC1) · 17 Nov 2016

Anonymous Referee #1 General Comments: R1.0. Saha and co-authors present a well-articulated investigation of the volatility properties of organic aerosol at two sites in the southeast United States. The opportunity at Centerville, AL during SOAS is uniquely favorable since there were so many collocated measurements (of meteorological metrics and AMS factors, for example). The authors do a good job motivating the juxtaposition of these data to other data they have collected in Raleigh, albeit at a different time of year. Overall, I find the goal of the work and its presentation quality to be acceptable for ACP and important for the scientific community. However, I have some concerns about the discussion. For example, I think the issue of solubility, while not specifically addressed by their measurements, should be better woven throughout their discussions in the manuscript to make it clearer that the properties of volatility

and solubility simultaneously affect the gas-particle partitioning of organic compounds. I look forward to discussing the following issues with the authors:

A1.0. We thank the reviewer for his/her review and useful comments. All of the items mentioned here are addressed in response to specific comments below.

R1.1. Page 2, Line 15: Although it is well-known that volatility plays a pivotal in gas-particle partitioning, I think it is worth mentioning that solubility will also be critical for this phenomenon, especially in places like the southeast US. Admittedly, solubility is outside the scope of this paper (and the RH of the observations is maintained relatively low at 30-40%). It is of course still vital and useful to gain an in-depth knowledge of the partitioning behavior of compounds to the "dry" organic phase. However, please also be careful throughout the text about statements like in line 17-18: "vapor pressure determines whether an organic compound is found in the particle- or gas-phase" since this is not exactly true for many compounds, which are highly volatile and highly soluble, for example. This should be reworded for completeness.

AR1.1. We thank the reviewer for pointing out the role of solubility in OA gas-particle partitioning. We certainly agree with the reviewer that solubility in water may also play an important role in the gas-particle partitioning of many organic species (e.g., isoprene related species). We have revised the text to address this point. "At equilibrium, volatility of organic species, specifically, saturation vapor pressure (or equivalently, saturation concentration, $C^*$; $\mu$g m-3) plays a vital role in determining their gas-particle partitioning (Donahue et al., 2006; Pankow, 1994). Solubility in water may also be critical for gas-particle partitioning for many species (Hennigan et al., 2009), especially in places with higher relative humidity, for example, in the southeast U.S. Enthalpies of vaporization ($\Delta$Hvap) dictate the change in partitioning with temperature (Donahue et al., 2006; Epstein et al., 2010). Although gas-particle partitioning is determined by the basic thermodynamic properties of OA species – their $C^*$, $\Delta$Hvap, and solubility– these, along with the impacts of non-ideal mixing on individual species, are generally unknown for ambient OA."

[Figure]

R1.2. Did the authors size select the particles before entry to the TD units? It does not appear so. Could they comment on their method of incorporating the size distribution information into their model? Did they use a moving sectional, fixed sectional, or a modal algorithm? Or was only a single diameter used as is implied on page 6, line 13? If this is the case, can the authors provide some insight into how the inaccuracies introduced in their model results from the width of the distributions (as seen in Fig. S6c,d)? Are there significant size-dependent particle losses in the system and are these temperature dependent?

AR1.2. We did not size select the particles before entry to the TD units. We heated a polydisperse distribution of particles in TDs and observed the changes in total volume/mass concentrations as a function of temperatures and residence times.

In our TD kinetic modeling, we assumed the volume median diameter (VMD) as a representative size for a polydisperse aerosol; a reasonable assumption to extract average properties of aerosol. This assumption has been examined in the past (Park et al., 2013) and shown to have a minuscule effect on modeled particle evaporation. For example, Park et al. (2013) tested this assumption (Fig. S4, SI in Park et al.) for a similar model formulation to ours and showed that assumptions of poly- and monodisperse size distributions to represent a single-mode aerosol lead to virtually indistinguishable model results. In our dual-TD method characterization paper (Saha et al., 2015), we included a sensitivity analysis (SI; section S.6) to explore the potential influence of changes in particle VMD within a generous range (which would have a much larger effect than simply including a polydisperse aerosol population). We have shown that sensitivity of VMD within this range does not substantially alter our derived volatility parameter values.

Regarding particle losses, we have applied empirically determined particle loss correction factors derived using non-volatile NaCl aerosol (this approach was discussed in Saha et al., 2015). Particle mass and number transmission through TDs is temperature dependent. The size dependence is much stronger for particle number transmission

than mass/volume transmission. This is because of the substantial diffusional loss of smaller size particles which contribute relatively little to the particle mass/volume concentrations. We conducted our NaCl loss characterization experiments with a size distribution that was broadly comparable to the ambient distribution. Therefore, correcting our data with the mass transmission factors derived from laboratory-generated NaCl aerosol introduces minimal additional uncertainty.

R1.3. Page 6, Line 19-21: what is the reason for putting only the fi sets that have been "accepted" against the TS-TD data through the VRT-TD analysis? What sets are undetected by not doing both applications for every set and then taking the best performers? On a related note, what initial fi values were used as input to the solver and how dependent was the "accepted" solution on these values? Were these initial conditions varied at all? Why are the authors confident they have evaluated the entire space of volatility distributions? AR1.3. For a given input temperature (T) and residence time (Rt), the evaporation kinetic model predicts a mass fraction remaining; MFR (T, Rt). In our method, the fitting can be done in one step (fitting all data from TS-TD and VRT-TD together) or two steps. We have found that the two-step fitting approach gives essentially identical results to fitting all data simultaneously. We, therefore, decided to use a two-step fitting approach because it narrows down the parameter space substantially in the first step, which reduces the computational requirements substantially. An additional advantage to applying the 'two-step' approach in this work is that it distinctly demonstrates the benefit of adding the additional dimension (Rt) to the traditional TD measurement space (T) via goodness of fit quantification across the [$\gamma$e, $\Delta$Hvap] space at each step (Fig. 3). Thus, this approach/presentation gives insight into the range of parameter values that can be used to explain observations from 'single-dimensional' perturbations (only T) in typical TD arrangements used in various past/ongoing studies.

For solving for fi distributions, we have used a non-linear constrained optimization solver (fmincon in Matlab). We tested our model with different initial guesses for fi distributions, and optimal solutions were found to be insensitive to the initial guess for

a given set of inputs. A constraint of $\Sigma f_i = 1$ was used. We have provided constraint for the lower ($f_i$ minimum = 0.02) and upper ($f_i$ maximum = 0.4) boundary for a $f_i$ value in each $C^*$ bin. This choice of a wide solution space for solving a $f_i$ value in each $C^*$ bin would address any sensitivity of the error to an optimum solution of $f_i$.

We have not evaluated the entire volatility distributions space using our method, and did not indicate this in the paper. However, we explicitly did mention that our selected $C^*$ bin range was based on our measurement conditions; specifically, the highest TD operating temperature and the average ambient OA loading provide limitations on the lower and upper $C^*$ bins we consider, respectively. Within this predefined range, our approach provides an empirical OA volatility distribution that explains the observed evaporation of bulk OA in our dual-TD system.

R1.4. Page 7, Line 8-9: How do the enthalpies of vaporization used in this WRF-Chem simulation compare with those derived from the observations here (could they add this to the methods section)? What are the implications of comparing the volatility distribution from a model to observations when they are using different enthalpies? Is this something that other researchers should take seriously when comparing model output to TD data? On Page 15, Lines 20-24, the authors provide some discussion of this issue. However, they seem to be assessing the sensitivity of the enthalpies in the model independently of the mass yields being used. Is this appropriate?

AR1.4. In our CTM evaluation, we focused on comparing our observed OA volatility distribution against the WRF/Chem output using the current treatment of OA in models. Therefore, we used the VBS parametrizations (e.g., $C^*$ bin range), SOA yields and $\Delta H_{vap}$ that are currently being used in the research community. For enthalpies of vaporization, the semi-empirical correlation by Epstein et al. (2010) ($\Delta H_{vap, i} = 130 - 11\log_{10}C^*_{i,298}$) was used in the WRF/Chem simulation (discussed in section 2.5). Our TD parameter fitting used a generalized functional form for $\Delta H_{vap}$ ($\Delta H_{vap, i} =$ intercept-slope ($\log_{10}C^*_{i,298}$)), where intercept and slope were fit parameters. We found that a $\Delta H_{vap, i} = 100 - 0$ ($\log_{10}C^*_{i,298}$) relationship 'best' explained the observed

temperature sensitivity of bulk OA in our TDs (See Fig. 3 and discussion in section 3.3).

It should be noted here that the reference VBS temperature was 25°C for both WRF/Chem run and TD fits. The difference in ΔHvap used in WRF/Chem runs and our TD-derived value would not have a significant effect on the comparison shown in Fig.10. This is because the modeled-measured OA volatility comparison was made at temperatures (SOAS campaign average T= 24.7°C; WRF/Chem simulated campaign average T= 23.8°C @ 2 m) that are very close to the VBS reference temperature (25°C). Murphy et al. (2011) also reported a low sensitivity of ΔHvap when predicting surface OA loading during the FAME-08 study using a 2D-VBS framework. However, the effect of ΔHvap could be very significant when simulating OA loading at high altitudes. Therefore, we recommend that researchers should take the influence of ΔHvap seriously, especially when comparing an OA volatility distribution from a CTM at a temperature that is very different from the reference temperature of VBS. We included relevant discussions in our revised manuscript in page 16, Lines 15-20.

"The difference in ΔHvap values used in WRF/Chem and our TD-derived values should not have a significant effect on the comparison shown in Fig.10. This is because the modeled-measured OA volatility comparison was made at temperatures (SOAS campaign average T= 24.7°C; WRF/Chem simulated campaign average T= 23.8°C) very close to the VBS reference temperature (25°C). Murphy et al. (2011) also reported a low sensitivity of ΔHvap when predicting surface OA loading during the FAME-08 study using a 2D-VBS framework. However, the effect of ΔHvap could be significant when simulating OA loading at low ambient temperatures and high altitudes."

Ideally, yields parameterization and ΔHvap should be internally consistent for a CTM input. Traditionally yields parameterization and ΔHvap have not been coupled and constrained consistently. Typically a ΔHvap of 30 - 40 KJ mol-1 has been assumed for atmospheric modeling. A derivation of consistent parameterizations for yields and ΔHvap as CTM inputs are not within the scope of our paper. Following previous work (Farina et al., 2010; Murphy et al., 2011), we briefly discussed the sensitivity of ΔHvap

to provide some insights into the influence of ΔHvap in a qualitative sense.

R1.5. Page 9, Line 8-9: I'm not familiar with the term "condensation sink diameter". Could the authors please explain it and potentially provide an equation in the supporting info for this quantity? How is it related to the more common term, condensation sink which is in units of inverse time?

AR1.5. Yes, the concept of condensation sink diameter ($d\_cs$) is related to condensation sink (CS). This concept is first described in Lehtinen et al. (2003). According to Lehtinen et al., "The condensation sink diameter of a distribution of particles with total number concentration N_tot, is the diameter where a monodisperse population of particles of number concentration N_tot should be placed to obtain the same total condensation sink (CS) as for the polydisperse distribution of interest." Mathematically, $2\pi D d\_cs\ F(d\_cs)\ N\_tot = 2\pi D(dp,i)(dp,i)(N\_i) = CS$ Where, D is the diffusion coefficient, F is the Fuchs and Sutugin correction factor, N_i is the number concentration of particles in size bin of dp,i. A new section (Sec. S2) has been added to the SI detailing the estimation of condensation sink diameter.

R1.6. Do the authors have an explanation for why the 80-0 and 80-4 cases with evaporation coefficient equal to 0.5 and 1.0 performed acceptably for the Raleigh cases and not for the Centerville cases? How close were they to being accepted with the VRT-TD data? It perhaps appears that they performed better for the TS-TD data at Centerville than they did at Raleigh (or at least the cases around them with discernible colors did).

AR1.6. Although the observed campaign-average evaporation in both data sets were indistinguishable at higher TD temperatures, there was a slight difference in evaporation profiles at various Rt at lower temperatures (see Figs 2.b and c). This slight differences in the observed evaporation profiles at 60 and 90 °C as shown in Figs. 2b and c are likely the cause of the differences between optimal parameter sets. We cannot comment on the factors leading to this difference, as our estimated volatility parameter values are empirical estimates that optimally describe bulk volatility properties

in combination with the assumed values of other parameters (D, $\sigma$, , MW).

R1.7. Page 10, Lines 5-10: The low enthalpies used in models are based on observation data (e.g. Offenberg et al. 2006; Pathak et al. 2007; Stanier et al., 2008) of so-called "effective" enthalpies, so please consider mentioning this for completeness; Stanier et al. (2008) were a bit higher than others and in line with the lower bound explored in this study. It has been argued that low enthalpies of vaporization result from describing systems with too few volatility surrogates (Riipinen et al., 2010). Could the authors comment on what they attribute the void between their high enthalpies and historical low enthalpies to? If they run their model with only two surrogates, for example, do they get low enthalpies as well?

AR1.7. We thank the reviewers for the suggestion. We added these references in our paper. As the reviewer correctly pointed out, these are not real enthalpies, but "effective" ones. The reported "effective" enthalpies are often much lower than enthalpies of chemical compounds relevant to atmospheric aerosols. By reducing the number of surrogate compounds, the system moves further away from realistic enthalpies. As a mixture evaporates, it is progressively enriched in less volatile compounds, slowing down evaporation in terms of the total mixture mass. Thus, the apparent sensitivity of the aerosol mass to temperature appears to be low. This translates to a low enthalpy of vaporization if one uses, for example, one surrogate compound. We have not tested the dependence of the apparent $\Delta$Hvap values on the number of surrogate compounds, as our purpose was to describe the aerosol using the VBS representation with a commonly used number of bins.

R1.8. Can the authors please provide some statistics to go along with the comparison in Fig. 4? For example, it would be useful to have mean bias, correlation coefficient and root mean square error so that future studies would have something succinct to use as a benchmark.

AR1.8. We have added values of coefficient of determination (r2) and root mean square

error (RMSE) for the measured and modeled MFRs in Fig 4.a and Fig.S9.

R1.9. In Fig. 4a, is the COA that of the ambient (unheated) sample? There does appear to be systematic overprediction for low COA cases compared to high COA cases. Is this true? Could the authors provide some statistics stratified by COA to assess this? If there is a relationship, could they comment on why it emerges from their approach?

AR1.9. Yes, the COA in Fig.4a is the measured ambient OA concentrations. We explored the relationship between COA and extracted volatility. Fig.AR.1 shows the scatter plot of (a) mean C* vs. ambient COA and (b) C*eff vs. ambient COA (this Fig is included in SI as Fig.S12). In a few low COA instances, the mean C* was found to be higher, but this trend is not consistent. The relative contribution of MO-OOA (more-oxidized oxygenated-OA) in COA in many of these instances was low (yellow/orange points in Fig. AR.1a, which likely influences this observation. However, there is large amounts of scatter in mean C* in the lowest COA range, so a consistent relationship is not evident. The C*eff vs. ambient COA plot shows an increasing trend of C*eff with COA. This is because the C*eff only considers the particle-phase components and was estimated following equilibrium partitioning theory – therefore this mild increase of C*eff is consistent with increasing partitioning of semi-volatile species to the particle-phase with increased COA.

R1.10. Page 13, Line 8: Please avoid using the word "relatively" and opt instead for a quantification of the difference between the afternoon and early morning.

AR1.10. Revised text. "OA appeared less volatile in the afternoon than early morning for both sites (Centreville: campaign average (Cˆ* ) ÌĚ ($\mu$g m-3) in the morning $\sim$ 0.25; afternoon $\sim$ 0.13 and Raleigh: morning $\sim$ 0.2; afternoon $\sim$ 0.12)."

R1.11. Sections 3.4-3.5: The authors have defined both mean(C*) and C*eff and chose the former for their analysis of time series and diurnal profiles. I think this may have been an unfortunate choice. This value, as they say, accounts for both particle- and

gas-phase compounds. It is not very surprising to me that it would be a more stable quantity, throughout the SOAS campaign at least. I wonder if they would capture the trends they are after better by using the C*eff which just accounts for the particle-phase material. This at least would seem to make more sense for correlating with AMS factors (Isoprene-OA, LO-OOA, and MO-OOA) since those apply just to the particle phase and not the total organic burden. Using this alternative approach would theoretically introduce a temperature and dilution dependence that might result in a more striking and meaningful variability. Of course, it may on the other hand yield a rather invariant trend similar to what they have already shown.

AR1.11. According to our definition, the mean C* is a log-mean of the volatility bins of organic species (particles + vapor), which gives a description of where the mass (center) of different volatility compounds is located. Therefore, the mean C* is a simpler representation of the volatility basis set (VBS). On the other hand, C*_eff is another simplified representation of OA volatility derived based on the Raoult's law and using arithmetic averaging. The C*_eff would indicate a current state of OA volatility and temperature- and dilution-dependence. We elected to use the mean C* for our extended analysis because we consider the log-mean a better representation of the volatility of ambient OA, not only what is measured in particle instruments.

We explored the correlation between different OA factors and bulk OA volatility using both metrics (mean C* and C*_eff). As an example, Fig. AR.2 shows the scatter plot of (a) mean C* vs. MO-OOA fraction and (b) C*_eff vs. MO-OOA fraction COA (this Fig. is now included in the SI as Fig.S13). While the exact correlations of course vary, the correlation coefficients and thus our general conclusions on the correlation between different OA factors and bulk OA volatility remain unchanged.

Minor Changes/Typos: R1.12. Pg 1, Line 11: Consider mentioning solubility as well. "quantitative estimates of the thermodynamic (volatility, water solubility, etc)"

AR1.12. We revised the text as suggested.

R1.13. Pg 1, Line 14: "one at a biogenic: : :"

AR1.13. We revised the text.

R1.14. Pg 2, Line 6-7: Of course SOA has also been shown to be introduced from oxidation of primary SVOCs and IVOCs followed by condensation and aqueous phase reactions of high soluble compounds. Depending on your definition of "secondary", it is also formed by heterogeneous oxidation of POA without evaporation. Please mention these here or don't be so specific about the VOCs role.

AR1.14. We revised the text. Page 2, Lines 5-6: "Secondary OA (SOA) is formed in the atmosphere via oxidation reactions of gas-phase organic species; it may also be formed by reactions in the particle (condensed) phase"

R1.15. Pg 5, Line 16-17: The VBS approach does not assume unity activity coefficients. Instead the activity coefficient are assumed to be lumped into the Cvap, making it C*. And so the activity coefficients have a value, but that value is assumed not to change from the lab or field to other mixtures or conditions.

AR1.15. We revised the text. Page5, Lines 22-23: "The VBS approach is based on an effective saturation concentration (C*) where the activity coefficient is assumed to be lumped into the saturation concentration."

R1.16. Page 6, Line 10: I disagree that the model is applied in an "inverse sense", as it appears that the authors are solving the evaporation problem in the forward way, brute-force, for all of their free parameter combinations and then comparing output to the observations. This could perhaps be labelled reverse engineering, but I think associating it with inverse modeling is inaccurate.

AR1.16. We used a non-linear constrained optimization solver ('fmincon' in Matlab) to extract OA volatility distribution by matching measured and modeled evaporation data. We revised the text. "The model is applied to extract OA properties such as the volatility distribution, $\Delta Hvap$, and $\gamma e$ as fitting parameters by matching measured and modeled

evaporation data."

R1.17. Page 6, Line 27: "common means to improve the performance of OA prediction in chemical transport models"

AR1.17. We revised the text as suggested.

R1.18. Fig. 2b,c: There is no explanation or legend for the colors of the trends. The text claims they distinguish different residence times but isn't this described by the x-axis? Please clarify this point.

AR1.18. Fig. 2a shows an OA MFR vs. temperature plot. A colour scale was used for Fig.2a to distinguish TD measurements data with different residence times across studies. It was mentioned in Fig.2 caption. The x-axis of Figs. 2b and c are residence times. Thus we do not need a residence time colour scale for these two panels. To clarify the legend issue, we added a sentence in Fig.2 caption: "Legend shown next to panel (a) applies to all panels (a-c)."

R1.19. Page 9, Line 14: Just write "volatility". Having the "/C*" is unnecessary. Page 15, Line 7 as well. Page 9 and Fig. 3: Neither the description in the text nor the figure caption should claim that the figure depicts the fi values. As far as I can tell, it does not.

AR1.19. We revised the text as suggested. We agree with the reviewer that Fig.3 visually does provide information about our fitted fi distributions. To clarify it we revised Fig.3 caption as, "Extraction process of OA gas-particle partitioning parameter ($\Delta$Hvap, $\gamma$e and fi) values. A fi distribution was solved for each combination of ($\Delta$Hvap, $\gamma$e) via evaporation kinetic model fits to campaign-average dual-TD observations"

R1.20. Page 9, Line 18: Please indicate here, and throughout the text, that this C* is at Tref (presumably 298 or 300 K). This should especially be made clear for any references to mean C*.

AR1.20. We revised the text as suggested. "A reference temperature (Tref) of 298 K is assumed. Any C* value reported in this paper should be considered at 298 K, unless

otherwise specified."

R1.21. Page 10 and Fig. 2: The notation for Ceff is not consistent with Csat_eff.

AR1.21. We revised the text as suggested.

R1.22. Page 13, Line 26-27: You do not really need this final sentence since the section you refer to comes directly next.

AR1.22. We revised the text as suggested.

R1.23. Page 16, Line 16: Please be quantitative rather than saying "Relatively less volatile".

AR1.23. We removed the word 'relatively'. Quantitative data is stated in page 13, Lines 16-18; "OA appeared less volatile in the afternoon than early in the morning for both sites (Centreville: campaign average (meanC*; $\mu$g m-3) in the morning $\sim$ 0.25; afternoon $\sim$ 0.13 and Raleigh: morning $\sim$ 0.2; afternoon $\sim$ 0.12)"

R1.24. Page 16, Line 30: Murphy et al. (2011) compared predictions with the 2D-VBS in a Lagrangian column CTM against the FAME data reported by Lee et al.

We thank the reviewer for the information. We cut that sentence and added the Murphy et al. (2011) reference.

References: Cappa, C. D.: A model of aerosol evaporation kinetics in a thermodenuder, Atmos. Meas. Tech., 3(3), 579–592, doi:10.5194/amt-3-579-2010, 2010.

Farina, S. C., Adams, P. J. and Pandis, S. N.: Modeling global secondary organic aerosol formation and processing with the volatility basis set: Implications for anthropogenic secondary organic aerosol, J. Geophys. Res. Atmospheres, 115(D9), D09202, doi:10.1029/2009JD013046, 2010.

Hennigan, C. J., Bergin, M. H., Russell, A. G., Nenes, A. and Weber, R. J.: Gas/particle partitioning of water-soluble organic aerosol in Atlanta, Atmos Chem Phys, 9(11),

3613–3628, doi:10.5194/acp-9-3613-2009, 2009.

Lee, B. H., Kostenidou, E., Hildebrandt, L., Riipinen, I., Engelhart, G. J., Mohr, C., DeCarlo, P. F., Mihalopoulos, N., Prevot, A. S. H., Baltensperger, U. and Pandis, S. N.: Measurement of the ambient organic aerosol volatility distribution: application during the Finokalia Aerosol Measurement Experiment (FAME-2008), Atmos Chem Phys, 10(24), 12149–12160, doi:10.5194/acp-10-12149-2010, 2010.

Lehtinen, K., Korhonen, H., Maso, M. D. and Kulmala, M.: On the concept of condensation sink diameter, Boreal Env. Res, 8, 405–411, 2003.

May, A. A., Levin, E. J. T., Hennigan, C. J., Riipinen, I., Lee, T., Collett, J. L., Jimenez, J. L., Kreidenweis, S. M. and Robinson, A. L.: Gas-particle partitioning of primary organic aerosol emissions: 3. Biomass burning, J. Geophys. Res. Atmospheres, 118(19), 11,327–11,338, doi:10.1002/jgrd.50828, 2013.

Murphy, B. N., Donahue, N. M., Fountoukis, C. and Pandis, S. N.: Simulating the oxygen content of ambient organic aerosol with the 2D volatility basis set, Atmos Chem Phys, 11(15), 7859–7873, doi:10.5194/acp-11-7859-2011, 2011.

Offenberg, J. H., T. E. Kleindienst, M. Jaoui, M. Lewandowski, and E. O. Edney (2006), Thermal properties of secondary organic aerosols, Geophys. Res. Lett., 33, L03816, doi:10.1029/2005GL024623.

Park, S. H., Rogak, S. N. and Grieshop, A. P.: A Two-Dimensional Laminar Flow Model for Thermodenuders Applied to Vapor Pressure Measurements, Aerosol Sci. Technol., 47(3), 283–293, doi:10.1080/02786826.2012.750711, 2013.

Ranjan, M., Presto, A. A., May, A. A. and Robinson, A. L.: Temperature Dependence of Gas–Particle Partitioning of Primary Organic Aerosol Emissions from a Small Diesel Engine, Aerosol Sci. Technol., 46(1), 13–21, doi:10.1080/02786826.2011.602761, 2012.

Riipinen, I., Pierce, J. R., Donahue, N. M. and Pandis, S. N.: Equilibrium time scales of

organic aerosol inside thermodenuders: Evaporation kinetics versus thermodynamics, Atmos. Environ., 44(5), 597–607, doi:10.1016/j.atmosenv.2009.11.022, 2010.

Saha, P. K., Khlystov, A. and Grieshop, A. P.: Determining Aerosol Volatility Parameters Using a "Dual Thermodenuder" System: Application to Laboratory-Generated Organic Aerosols, Aerosol Sci. Technol., 49(8), 620–632, doi:10.1080/02786826.2015.1056769, 2015.

Saleh, R., Walker, J. and Khlystov, A.: Determination of saturation pressure and enthalpy of vaporization of semi-volatile aerosols: The integrated volume method, J. Aerosol Sci., 39(10), 876–887, doi:10.1016/j.jaerosci.2008.06.004, 2008.

Stanier, Donahue, and Pandis (2008), Parameterization of secondary organic aerosol mass fractions from smog chamber data, Atmos. Environ. 42, 10, 2276-2299, doi:10.1016/j.atmosenv.20

Please also note the supplement to this comment:
http://www.atmos-chem-phys-discuss.net/acp-2016-575/acp-2016-575-AC1-supplement.pdf

[Figure]

[Figure]

**Fig. 1.** (Fig.AR.1) Scatter plot of (a) mean C* vs. ambient OA loading (COA); (b) C*eff vs. ambient OA loading (COA). Results are shown from the Centreville campaign.

[Figure]

**Fig. 2.** (Fig.AR.2) Scatter plot of (a) mean C* vs. MO-OOA fraction in COA; (b) C*eff vs. MO-OOA fraction in COA. Results are shown from the Centreville campaign.

**Supplement:**

Response to reviewers comments for the paper "**Quantifying the volatility of organic aerosol in the southeastern U.S.**" by Provat K. Saha et al.

To Whom It May Concern:

We would like to thank the two reviewers for their thoughtful and helpful comments. In addressing these comments, we feel we have substantially improved the manuscript. Here we provide a point-by-point response to review comments.

Please find below detailed responses in normal text (with direct quotes from the revised manuscript shown in *italics*) to the comments and suggestions (shown in **blue text**) offered by the two reviewers.

Best regards,

The Authors

Anonymous Referee #1

General Comments:

**R1.0.** Saha and co-authors present a well-articulated investigation of the volatility properties of organic aerosol at two sites in the southeast United States. The opportunity at Centerville, AL during SOAS is uniquely favorable since there were so many collocated measurements (of meteorological metrics and AMS factors, for example). The authors do a good job motivating the juxtaposition of these data to other data they have collected in Raleigh, albeit at a different time of year. Overall, I find the goal of the work and its presentation quality to be acceptable for ACP and important for the scientific community. However, I have some concerns about the discussion. For example, I think the issue of solubility, while not specifically addressed by their measurements, should be better woven throughout their discussions in the manuscript to make it clearer that the properties of volatility and solubility simultaneously affect the gas-particle partitioning of organic compounds. I look forward to discussing the following issues with the authors:

**A1.0.** We thank the reviewer for his/her review and useful comments. All of the items mentioned here are addressed in response to specific comments below.

**R1.1.** Page 2, Line 15: Although it is well-known that volatility plays a pivotal in gasparticle partitioning, I think it is worth mentioning that solubility will also be critical for this phenomenon, especially in places like the southeast US. Admittedly, solubility is outside the scope of this paper (and the RH of the observations is maintained relatively low at 30-40%). It is of course still vital and useful to gain an in-depth knowledge of the partitioning behavior of compounds to the "dry" organic phase. However, please also be careful throughout the text about statements like in line 17-18: "vapor pressure determines whether an organic compound is found in the particle- or gas-phase" since this is not exactly true for many compounds, which are highly volatile and highly soluble, for example. This should be reworded for completeness.

**AR1.1.** We thank the reviewer for pointing out the role of solubility in OA gas-particle partitioning. We certainly agree with the reviewer that solubility in water may also play an important role in the gas-particle partitioning of many organic species (e.g., isoprene related species). We have revised the text to address this point.

  *"At equilibrium, volatility of organic species, specifically, saturation vapor pressure (or equivalently, saturation concentration, $C^*$; $\mu g\ m^{-3}$) plays a vital role in determining their gas-particle partitioning (Donahue et al., 2006; Pankow, 1994). Solubility in water may also be critical for gas-particle partitioning for many species (Hennigan et al., 2009), especially in places with higher relative humidity, for example, in the southeast U.S. Enthalpies of vaporization ($\Delta H_{vap}$) dictate the change in partitioning with temperature (Donahue et al., 2006; Epstein et al., 2010). Although gas-particle partitioning is determined by the basic thermodynamic properties of OA species – their $C^*$, $\Delta H_{vap}$, and solubility– these, along with the impacts of non-ideal mixing on individual species, are generally unknown for ambient OA."*

**R1.2.** Did the authors size select the particles before entry to the TD units? It does not appear so. Could they comment on their method of incorporating the size distribution information into their model? Did they use a moving sectional, fixed sectional, or a modal algorithm? Or was only a

single diameter used as is implied on page 6, line 13? If this is the case, can the authors provide some insight into how the inaccuracies introduced in their model results from the width of the distributions (as seen in Fig. S6c,d)? Are there significant size-dependent particle losses in the system and are these temperature dependent?

**AR1.2.** We did not size select the particles before entry to the TD units. We heated a polydisperse distribution of particles in TDs and observed the changes in total volume/mass concentrations as a function of temperatures and residence times.

In our TD kinetic modeling, we assumed the volume median diameter (VMD) as a representative size for a polydisperse aerosol; a reasonable assumption to extract average properties of aerosol. This assumption has been examined in the past (Park et al., 2013) and shown to have a minuscule effect on modeled particle evaporation. For example, Park et al. (2013) tested this assumption (Fig. S4, SI in Park et al.) for a similar model formulation to ours and showed that assumptions of poly- and monodisperse size distributions to represent a single-mode aerosol lead to virtually indistinguishable model results. In our dual-TD method characterization paper (Saha et al., 2015), we included a sensitivity analysis (SI; section S.6) to explore the potential influence of changes in particle VMD within a generous range (which would have a much larger effect than simply including a polydisperse aerosol population). We have shown that sensitivity of VMD within this range does not substantially alter our derived volatility parameter values.

Regarding particle losses, we have applied empirically determined particle loss correction factors derived using non-volatile NaCl aerosol (this approach was discussed in Saha et al., 2015). Particle mass and number transmission through TDs is temperature dependent. The size dependence is much stronger for particle number transmission than mass/volume transmission. This is because of the substantial diffusional loss of smaller size particles which contribute relatively little to the particle mass/volume concentrations. We conducted our NaCl loss characterization experiments with a size distribution that was broadly comparable to the ambient distribution. Therefore, correcting our data with the mass transmission factors derived from laboratory-generated NaCl aerosol introduces minimal additional uncertainty.

**R1.3.** Page 6, Line 19-21: what is the reason for putting only the fi sets that have been "accepted" against the TS-TD data through the VRT-TD analysis? What sets are undetected by not doing both applications for every set and then taking the best performers? On a related note, what initial fi values were used as input to the solver and how dependent was the "accepted" solution on these values? Were these initial conditions varied at all? Why are the authors confident they have evaluated the entire space of volatility distributions?

**AR1.3.** For a given input temperature (T) and residence time (Rt), the evaporation kinetic model predicts a mass fraction remaining; MFR (T, Rt). In our method, the fitting can be done in one step (fitting all data from TS-TD and VRT-TD together) or two steps. We have found that the two-step fitting approach gives essentially identical results to fitting all data simultaneously. We, therefore, decided to use a two-step fitting approach because it narrows down the parameter space substantially in the first step, which reduces the computational requirements substantially. An additional advantage to applying the 'two-step' approach in this work is that it distinctly demonstrates the benefit of adding the additional dimension (Rt) to the traditional TD measurement space (T) via goodness of fit quantification across the $[\gamma_e, \Delta H_{vap}]$ space at each step (Fig. 3). Thus, this approach/presentation gives insight into the range of parameter values that can be used to explain observations from 'single-dimensional' perturbations (only T) in typical TD arrangements used in various past/ongoing studies.

For solving for $f_i$ distributions, we have used a non-linear constrained optimization solver (*fmincon* in Matlab). We tested our model with different initial guesses for $f_i$ distributions, and optimal solutions were found to be insensitive to the initial guess for a given set of inputs. A constraint of $\Sigma f_i = 1$ was used. We have provided constraint for the lower ($f_i$ minimum = 0.02) and upper ($f_i$ maximum = 0.4) boundary for a $f_i$ value in each $C^*$ bin. This choice of a wide solution space for solving a $f_i$ value in each $C^*$ bin would address any sensitivity of the error to an optimum solution of $f_i$.

We have not evaluated the entire volatility distributions space using our method, and did not indicate this in the paper. However, we explicitly did mention that our selected $C^*$ bin range was based on our measurement conditions; specifically, the highest TD operating temperature and the average ambient OA loading provide limitations on the lower and upper C* bins we consider,

respectively. Within this predefined range, our approach provides an empirical OA volatility distribution that explains the observed evaporation of bulk OA in our dual-TD system.

**R1.4.** Page 7, Line 8-9: How do the enthalpies of vaporization used in this WRF-Chem simulation compare with those derived from the observations here (could they add this to the methods section)? What are the implications of comparing the volatility distribution from a model to observations when they are using different enthalpies? Is this something that other researchers should take seriously when comparing model output to TD data? On Page 15, Lines 20-24, the authors provide some discussion of this issue. However, they seem to be assessing the sensitivity of the enthalpies in the model independently of the mass yields being used. Is this appropriate?

**AR1.4.** In our CTM evaluation, we focused on comparing our observed OA volatility distribution against the WRF/Chem output using the current treatment of OA in models. Therefore, we used the VBS parametrizations (e.g., C* bin range), SOA yields and $\Delta H_{vap}$ that are currently being used in the research community. For enthalpies of vaporization, the semi-empirical correlation by Epstein et al. (2010) ($\Delta H_{vap, i} = 130-11\log10C^*_{i,298}$) was used in the WRF/Chem simulation (discussed in section 2.5). Our TD parameter fitting used a generalized functional form for $\Delta H_{vap}$ ($\Delta H_{vap, i} = $ intercept-slope ($\log10C^*_{i,298}$)), where intercept and slope were fit parameters. We found that a $\Delta H_{vap, i} = 100-0$ ($\log10C^*_{i,298}$) relationship 'best' explained the observed temperature sensitivity of bulk OA in our TDs (See Fig. 3 and discussion in section 3.3).

It should be noted here that the reference VBS temperature was 25°C for both WRF/Chem run and TD fits. The difference in $\Delta H_{vap}$ used in WRF/Chem runs and our TD-derived value would not have a significant effect on the comparison shown in Fig.10. This is because the modeled-measured OA volatility comparison was made at temperatures (SOAS campaign average T= 24.7°C; WRF/Chem simulated campaign average T= 23.8°C @ 2 m) that are very close to the VBS reference temperature (25°C). Murphy et al. (2011) also reported a low sensitivity of $\Delta H_{vap}$ when predicting surface OA loading during the FAME-08 study using a 2D-VBS framework. However, the effect of $\Delta H_{vap}$ could be very significant when simulating OA loading at high altitudes. Therefore, we recommend that researchers should take the influence of $\Delta H_{vap}$

seriously, especially when comparing an OA volatility distribution from a CTM at a temperature that is very different from the reference temperature of VBS. We included relevant discussions in our revised manuscript in page 16, Lines 15-20.

*"The difference in $\Delta H_{vap}$ values used in WRF/Chem and our TD-derived values should not have a significant effect on the comparison shown in Fig.10. This is because the modeled-measured OA volatility comparison was made at temperatures (SOAS campaign average T= 24.7°C; WRF/Chem simulated campaign average T= 23.8°C) very close to the VBS reference temperature (25°C). Murphy et al. (2011) also reported a low sensitivity of $\Delta H_{vap}$ when predicating surface OA loading during the FAME-08 study using a 2D-VBS framework. However, the effect of $\Delta H_{vap}$ could be significant when simulating OA loading at low ambient temperatures and high altitudes."*

Ideally, yields parameterization and $\Delta H_{vap}$ should be internally consistent for a CTM input. Traditionally yields parameterization and $\Delta H_{vap}$ have not been coupled and constrained consistently. Typically a $\Delta H_{vap}$ of 30 - 40 KJ mol$^{-1}$ has been assumed for atmospheric modeling. A derivation of consistent parameterizations for yields and $\Delta H_{vap}$ as CTM inputs are not within the scope of our paper. Following previous work (Farina et al., 2010; Murphy et al., 2011), we briefly discussed the sensitivity of $\Delta H_{vap}$ to provide some insights into the influence of $\Delta H_{vap}$ in a qualitative sense.

**R1.5.** Page 9, Line 8-9: I'm not familiar with the term "condensation sink diameter". Could the authors please explain it and potentially provide an equation in the supporting info for this quantity? How is it related to the more common term, condensation sink which is in units of inverse time?

**AR1.5.** Yes, the concept of condensation sink diameter ($d_{cs}$) is related to condensation sink (CS). This concept is first described in Lehtinen et al. (2003). According to Lehtinen et al.,

"The condensation sink diameter of a distribution of particles with total number concentration $N_{tot}$, is the diameter where a monodisperse population of particles of number concentration $N_{tot}$ should be placed to obtain the same total condensation sink (CS) as for the polydisperse distribution of interest."

Mathematically, $2\pi D d_{cs} F(d_{cs}) N_{tot} = 2\pi D \sum F(d_{p,i}) d_{p,i} N_i = CS$

Where, D is the diffusion coefficient, F is the Fuchs and Sutugin correction factor, $N_i$ is the number concentration of particles in size bin of $d_{p,i}$. A new section (Sec. S2) has been added to the SI detailing the estimation of condensation sink diameter.

**R1.6.** Do the authors have an explanation for why the 80-0 and 80-4 cases with evaporation coefficient equal to 0.5 and 1.0 performed acceptably for the Raleigh cases and not for the Centerville cases? How close were they to being accepted with the VRT-TD data? It perhaps appears that they performed better for the TS-TD data at Centerville than they did at Raleigh (or at least the cases around them with discernible colors did).

**AR1.6.** Although the observed campaign-average evaporation in both data sets were indistinguishable at higher TD temperatures, there was a slight difference in evaporation profiles at various Rt at lower temperatures (see Figs 2.b and c). This slight differences in the observed evaporation profiles at 60 and 90 °C as shown in Figs. 2b and c are likely the cause of the differences between optimal parameter sets. We cannot comment on the factors leading to this difference, as our estimated volatility parameter values are empirical estimates that optimally describe bulk volatility properties in combination with the assumed values of other parameters (D, *σ, ρ, MW)*.

**R1.7.** Page 10, Lines 5-10: The low enthalpies used in models are based on observation data (e.g. Offenberg et al. 2006; Pathak et al. 2007; Stanier et al., 2008) of so-called "effective" enthalpies, so please consider mentioning this for completeness; Stanier et al. (2008) were a bit higher than others and in line with the lower bound explored in this study. It has been argued that low enthalpies of vaporization result from describing systems with too few volatility surrogates (Riipinen et al., 2010). Could the authors comment on what they attribute the void between their high enthalpies and historical low enthalpies to? If they run their model with only two surrogates, for example, do they get low enthalpies as well?

**AR1.7.** We thank the reviewers for the suggestion. We added these references in our paper. As the reviewer correctly pointed out, these are not real enthalpies, but "effective" ones. The reported "effective" enthalpies are often much lower than enthalpies of chemical compounds relevant to atmospheric aerosols. By reducing the number of surrogate compounds, the system

moves further away from realistic enthalpies. As a mixture evaporates, it is progressively enriched in less volatile compounds, slowing down evaporation in terms of the total mixture mass. Thus, the apparent sensitivity of the aerosol mass to temperature appears to be low. This translates to a low enthalpy of vaporization if one uses, for example, one surrogate compound. We have not tested the dependence of the apparent $\Delta H_{vap}$ values on the number of surrogate compounds, as our purpose was to describe the aerosol using the VBS representation with a commonly used number of bins.

**R1.8.** Can the authors please provide some statistics to go along with the comparison in Fig. 4? For example, it would be useful to have mean bias, correlation coefficient and root mean square error so that future studies would have something succinct to use as a benchmark.

**AR1.8.** We have added values of coefficient of determination ($r^2$) and root mean square error (RMSE) for the measured and modeled MFRs in Fig 4.a and Fig.S9.

**R1.9.** In Fig. 4a, is the COA that of the ambient (unheated) sample? There does appear to be systematic overprediction for low COA cases compared to high COA cases. Is this true? Could the authors provide some statistics stratified by COA to assess this? If there is a relationship, could they comment on why it emerges from their approach?

**AR1.9.** Yes, the $C_{OA}$ in Fig.4a is the measured ambient OA concentrations.

We explored the relationship between $C_{OA}$ and extracted volatility. Fig.AR.1 shows the scatter plot of (a) mean $C^*$ vs. ambient $C_{OA}$ and (b) $C^*_{eff}$ vs. ambient $C_{OA}$ (this Fig. is included in SI as Fig.S12). In a few low $C_{OA}$ instances, the mean $C^*$ was found to be higher, but this trend is not consistent. The relative contribution of MO-OOA (more-oxidized oxygenated-OA) in $C_{OA}$ in many of these instances was low (yellow/orange points in Fig. AR.1a, which likely influences this observation. However, there is large amounts of scatter in mean $C^*$ in the lowest $C_{OA}$ range, so a consistent relationship is not evident. The $C^*_{eff}$ vs. ambient $C_{OA}$ plot shows an increasing trend of $C^*_{eff}$ with $C_{OA}$. This is because the $C^*_{eff}$ only considers the particle-phase components and was estimated following equilibrium partitioning theory – therefore this mild increase of $C^*_{eff}$ is consistent with increasing partitioning of semi-volatile species to the particle-phase with increased $C_{OA}$.

[Figure]

Fig.AR.1: Scatter plot of (a) mean C* vs. ambient OA loading (C$_{OA}$); (b) C*$_{eff}$ vs. ambient OA loading (C$_{OA}$). Results are shown from the Centreville campaign. Fig.S12

**R1.10.** Page 13, Line 8: Please avoid using the word "relatively" and opt instead for a quantification of the difference between the afternoon and early morning.

**AR1.10.** Revised text.

 "*OA appeared less volatile in the afternoon than early morning for both sites (Centreville: campaign average $\overline{C}^*$ (µg m$^{-3}$) in the morning ~ 0.25; afternoon ~ 0.13 and Raleigh: morning ~ 0.2; afternoon ~ 0.12).*"

**R1.11.** Sections 3.4-3.5: The authors have defined both mean(C*) and C*eff and chose the former for their analysis of time series and diurnal profiles. I think this may have been an unfortunate choice. This value, as they say, accounts for both particle- and gas-phase compounds. It is not very surprising to me that it would be a more stable quantity, throughout the SOAS campaign at least. I wonder if they would capture the trends they are after better by using the C*eff which just accounts for the particle-phase material. This at least would seem to make more sense for correlating with AMS factors (Isoprene-OA, LO-OOA, and MO-OOA) since those apply just to the particle phase and not the total organic burden. Using this alternative approach would theoretically introduce a temperature and dilution dependence that might result in a more striking and meaningful variability. Of course, it may on the other hand yield a rather invariant trend similar to what they have already shown.

**AR1.11.** According to our definition, the mean C* is a log-mean of the volatility bins of organic species (particles + vapor) ($\overline{C^*} = 10^{\sum f_i \log_{10} C_i^*}$), which gives a description of where the mass (center) of different volatility compounds is located. Therefore, the mean C* is a simpler representation of the volatility basis set (VBS). On the other hand, C*_eff is another simplified representation of OA volatility ($C_{eff}^* = \sum x_i C_i^*$) derived based on the Raoult's law and using arithmetic averaging. The C*_eff would indicate a current state of OA volatility and temperature- and dilution-dependence. We elected to use the mean C* for our extended analysis because we consider the log-mean a better representation of the volatility of ambient OA, not only what is measured in particle instruments.

We explored the correlation between different OA factors and bulk OA volatility using both metrics (mean C* and C*_eff). As an example, Fig. AR.2 shows the scatter plot of (a) mean C* vs. MO-OOA fraction and (b) C*_eff vs. MO-OOA fraction C$_{OA}$ (this Fig. is now included in the SI as Fig.S13). While the exact correlations of course vary, the correlation coefficients and thus our general conclusions on the correlation between different OA factors and bulk OA volatility remain unchanged.

[Figure]

Fig.AR.2: Scatter plot of (a) mean C* vs. MO-OOA fraction in C$_{OA}$; (b) C*$_{eff}$ vs. MO-OOA fraction in C$_{OA}$. Results are shown from the Centreville campaign.

Minor Changes/Typos:

**R1.12.** Pg 1, Line 11: Consider mentioning solubility as well. "quantitative estimates of the thermodynamic (volatility, water solubility, etc)"

**AR1.12.** We revised the text as suggested.

**R1.13.** Pg 1, Line 14: "one at a biogenic: : :"

**AR1.13.** We revised the text

**R1.14.** Pg 2, Line 6-7: Of course SOA has also been shown to be introduced from oxidation of primary SVOCs and IVOCs followed by condensation and aqueous phase reactions of high soluble compounds. Depending on your definition of "secondary", it is also formed by heterogeneous oxidation of POA without evaporation. Please mention these here or don't be so specific about the VOCs role.

**AR1.14.** We revised the text. Page 2, Lines 5-6: *"Secondary OA (SOA) is formed in the atmosphere via oxidation reactions of gas-phase organic species; it may also be formed by reactions in the particle (condensed) phase"*

**R1.15.** Pg 5, Line 16-17: The VBS approach does not assume unity activity coefficients. Instead the activity coefficient are assumed to be lumped into the Cvap, making it C*. And so the activity coefficients have a value, but that value is assumed not to change from the lab or field to other mixtures or conditions.

**AR1.15.** We revised the text. Page5, Lines 22-23: *"The VBS approach is based on an effective saturation concentration (C*) where the activity coefficient is assumed to be lumped into the saturation concentration."*

**R1.16.** Page 6, Line 10: I disagree that the model is applied in an "inverse sense", as it appears that the authors are solving the evaporation problem in the forward way, brute-force, for all of their free parameter combinations and then comparing output to the observations. This could perhaps be labelled reverse engineering, but I think associating it with inverse modeling is inaccurate.

**AR1.16.** We used a non-linear constrained optimization solver ('*fmincon*' in Matlab) to extract OA volatility distribution by matching measured and modeled evaporation data.

We revised the text. *"The model is applied to extract OA properties such as the volatility distribution, $\Delta H_{vap}$, and $\gamma_e$ as fitting parameters by matching measured and modeled evaporation data."*

**R1.17.** Page 6, Line 27: "common means to improve the performance of OA prediction in chemical transport models"

**AR1.17.** We revised the text as suggested.

**R1.18.** Fig. 2b,c: There is no explanation or legend for the colors of the trends. The text claims they distinguish different residence times but isn't this described by the x-axis? Please clarify this point.

**AR1.18.** Fig. 2a shows an OA MFR vs. temperature plot. A colour scale was used for Fig.2a to distinguish TD measurements data with different residence times across studies. It was mentioned in Fig.2 caption. The x-axis of Figs. 2b and c are residence times. Thus we do not need a residence time colour scale for these two panels. To clarify the legend issue, we added a sentence in Fig.2 caption: "*Legend shown next to panel (a) applies to all panels (a-c).*"

**R1.19.** Page 9, Line 14: Just write "volatility". Having the "/C*" is unnecessary. Page 15, Line 7 as well. Page 9 and Fig. 3: Neither the description in the text nor the figure caption should claim that the figure depicts the fi values. As far as I can tell, it does not.

**AR1.19.** We revised the text as suggested.

We agree with the reviewer that Fig.3 visually does provide information about our fitted $f_i$ distributions. To clarify it we revised Fig.3 caption as, "*Extraction process of OA gas-particle partitioning parameter ($\Delta H_{vap}$, $\gamma_e$ and $f_i$) values. A $f_i$ distribution was solved for each combination of ($\Delta H_{vap}$, $\gamma_e$) via evaporation kinetic model fits to campaign-average dual-TD observations*"

**R1.20.** Page 9, Line 18: Please indicate here, and throughout the text, that this C* is at Tref (presumably 298 or 300 K). This should especially be made clear for any references to mean C*.

**AR1.20.** We revised the text as suggested. "*A reference temperature ($T_{ref}$) of 298 K is assumed. Any C* value reported in this paper should be considered at 298 K, unless otherwise specified.*"

**R1.21.** Page 10 and Fig. 2: The notation for Ceff is not consistent with Csat_eff.

**AR1.21.** We revised the text as suggested.

**R1.22.** Page 13, Line 26-27: You do not really need this final sentence since the section you refer to comes directly next.

**AR1.22.** We revised the text as suggested.

**R1.23.** Page 16, Line 16: Please be quantitative rather than saying "Relatively less volatile".

**AR1.23.** We removed the word 'relatively'. Quantitative data is stated in page 13, Lines 16-18; **"***OA appeared less volatile in the afternoon than early in the morning for both sites (Centreville: campaign average* $\overline{C^*}$ *(µg m$^{-3}$) in the morning ~ 0.25; afternoon ~ 0.13 and Raleigh: morning ~ 0.2; afternoon ~ 0.12)***"*

**R1.24.** Page 16, Line 30: Murphy et al. (2011) compared predictions with the 2D-VBS in a Lagrangian column CTM against the FAME data reported by Lee et al.

We thank the reviewer for the information. We cut that sentence and added the Murphy et al. (2011) reference.

Anonymous Referee #2

General Comments:

**R2.0.** This manuscripts presents measurements and analysis of the evaporation behavior of organic aerosol (OA) measured at two different locations – one more rural, one more urban, but both influenced by significant concentrations of BVOCs. Measurements were taken with a dual thermodenuder system in which both the temperature and residence time were varied. Main conclusions of the work include that the OA evaporation behavior was fairly similar at the two sites and did not vary much over the course of the measurement campaign, and also that much of the OA is in low-volatility bins which are currently not represented in (most) air-quality models. The manuscript is well written and generally well argued, and overall the topic and quality of the manuscript makes it suitable for publication in ACP. However, I have a few concerns about the analysis and interpretation as explained below which I suggest the authors should address before publication.

**A2.0.** We thank the reviewer for his/her review and useful comments. We address the specific comments below.

Major comments:

**R2.1.** The authors focus on the "best fit solution" and address to some extent the sensitivity of the error (SSR) to changes in dHvap and gamma (e.g. Fig. 3). I request that they also address the sensitivity of the error to changes in the fi. I am particularly worried that the fi in the lower C\* bins may not be well constrained by the data.

**AR2.1.** For solving a $f_i$ distribution, we have used a non-linear constrained optimization solver (*fmincon* in Matlab). We provided constraint for the lower ($f_i$ minimum = 0.02) and upper ($f_i$ maximum = 0.4) boundary for a $f_i$ value in each $C^*$ bin. This choice of a wide solution space for solving a $f_i$ value in each $C^*$ bin should address any sensitivity of the error to an optimum solution of $f_i$.

As we discussed in our manuscript that our selected $C^*$ bin range was based on our measurement conditions; specifically, the highest TD operating temperature and the average ambient OA loading provide limitations on the lower and upper C\* bins we consider, respectively.

Page 6, Lines 5-6, "*With the above $C^*$ bin limits, materials having $C^* < 10^{-4}$ μg m$^{-3}$ are lumped into the lowest bin……*"

Bins lower than ~$10^{-4}$ are not constrained in our data set because maximum TD operating temperature was 180°C. Materials those survived at 180 °C may well be even lower volatility, but we need to see it evaporate to be able to say anything quantitative about volatility. The key point is that within this predefined range, our approach provides an empirical OA volatility distribution that explains the observed evaporation of bulk OA in our dual-TD system.

**R2.2.** Regarding higher C\* bins, on page 6 lines 1-2 the stated reason for not including C\*>10 _g/m3 is that less than 5% of the materials would be present in the condensed phase at the average COA of 5 _g/m3. Common VBS bins used range from 0.1 to 100 _g/m3 for ambient OA concentration of _10 _g/m3. Elevated OA episodes (OA > 10 _g/m3) were observed, especially at the Raleigh site (Figure S7). It would be reasonable to include the 100 _g/m3 bin in the model to account for these episodes. Have the authors investigated the effects of including higher volatility bins on the fitted model parameters? As referee #1 pointed out and according to Riipinen et al. (2010), a two-surrogate product model could lead to very different conclusions about dHvap. How does the inclusion of more and lower volatility bins affect modeled dHvap? In other words, does bin selection introduce bias in to model results (and if so, how much)?

**AR2.2.** As noted in the manuscript and above (AR1.3), the selection of $C^*$ bin range in our fitting was based on our measurement conditions, specifically the highest TD operating temperature and average ambient OA loading. We performed sensitivity analysis on the selection of C* bin range. Fig. AR.3 shows an example result showing the comparison between fits with ranges of $C^* = [10^{-4}$ to $10\ \mu g\ m^{-3}]$ (base case used in our paper) versus $C^* = [10^{-3}$ to $10^2\ \mu g\ m^{-3}]$ of the Raleigh campaign average TD observations. Result indicates that $C^* = [10^{-3}$ to $10^2\ \mu g\ m^{-3}]$ fails to recreate the observed evaporations at higher TD temperature and that there is an indistinguishable change at lower temperature conditions (where the $C^* = 10^2\ \mu g\ m^{-3}$ would have any influence on the fit).

[Figure]

Fig.AR.3: Sensitivity of C* bin range in fitting TD data using an evaporation kinetic model. The best fitted curves with ($\Delta H_{vap} = 100\ kj\ mol^{-1}$ and $\gamma_e = 0.5$) are shown.

The reviewer correctly pointed out that there were a few elevated $C_{OA}$ episodes ($\geq 10\ \mu g\ m^{-3}$) during the Raleigh campaign. However, these episodes were for a relatively short period of time ( 2-3 hours) and the time required for collecting a complete set of thermograms using our TD setup was ~ 4-5 hours, we were not been able to capture a complete set of data under a consistently elevated $C_{OA}$ which could be fitted to constrain higher volatility bins.

As we discussed earlier in response to reviewer #1 (AR1.7) Comment, by reducing the number of surrogate compounds, the system representation would potentially move further away from realistic enthalpies. Therefore, we focused to describe the aerosol using the VBS representation

with a commonly-used number of bins. We have not tested the dependence of the apparent $\Delta H_{vap}$ values on the number of surrogate compounds. Our goal of this study was to describe ambient OA within this pre-defined framework. In our fitting, we represented $\Delta H_{vap}$ as a function of $C^*$ bin ($\Delta H_{vap}$ = intercept-slope ($\log_{10}C^*$)) and found that a relatively 'shallow' $\log_{10}C^*$ dependence (slope ~ 0, 4) better explain the observed temperature sensitivity of bulk OA in TDs. This suggests that an effective $\Delta H_{vap}$ required to explain the observed temperature sensitivity of bulk OA would be less sensitive to an inclusion of higher/lower C* bins within a realistic range. However, we do recommend that further work explores the sensitivity of the apparent $\Delta H_{vap}$ to the number of surrogate bins in future studies considering the use of basis sets with fewer bins.

**R2.3.** Page 6 Line 21-23: Have the authors attempted to apply the two models in a different order (VRT-TD first then TS-TD)? How much does this change the results?

**AR2.3.** The order of model application has no effect on derived parameters. We have used the same model for fitting data from TS-TD and VRT-TD. For a given input temperature (T) and residence time (Rt), the evaporation kinetics model predicts a mass fraction remaining; MFR (T, Rt). In our method, the fitting can be done in one step (fitting all data from TS-TD and VRT-TD together) or two steps. We have found that the two-step fitting approach gives identical results to fitting all data simultaneously. We, therefore, elected to use a two-step fitting approach because it narrows down the parameter space substantially in the first step, which reduces the computational requirements substantially. An additional advantage to applying the 'two-step' approach in this work is that it distinctly demonstrates the benefit of the additional dimension (Rt) we have added to the traditional TD measurement space (T) via goodness of fit quantification across the [$\gamma_e$, $\Delta H_{vap}$] space at each step (Fig. 3).

**R2.4.** Page 13 Line 1-6: mean C* is perhaps not a suitable metric to correlate with the fraction of isoprene OA. If isoprene OA was in fact much less volatile, its overall contribution to mean C* would be minor; the calculated mean C* value would be dominated by fi values of more volatile surrogate compounds. Using Ceff would present similar issues. It seems more appropriate to investigate the correlation between isoprene OA and individual bin fi's.

**A2.4.** We thank the reviewer for this suggestion. We have performed this additional analysis and a summary result is shown below:

| $\log_{10}C^*$ bin | -4 | -3 | -2 | -1 | 0 | 10 | mean $C^*$ | $C^*$_eff |
|---|---|---|---|---|---|---|---|---|
| Pearson R value (Iso-OA fraction vs $f_i$'s) | 0.02 | 0.29 | -0.07 | -0.06 | -0.14 | 0.04 | -0.06 | 0.19 |

Based on our analysis we did not find any statistically-significant relationship between isoprene-OA fraction and $f_i$'s in any individual $C^*$ bin. This result is consistent with our original observation of no correlation between mean C* of bulk OA and the fractional contribution of isoprene-OA to $C_{OA}$. This new result is included in Table S2 and discussed on Page 13, Lines 29-30: *"Neither were statistically-significant relationships found between the isoprene-OA fraction and $f_i$'s in any particular $C^*$ bin (see Table S2)."*

**R2.5.** Page 6, line 7: Was a constant collection efficiency of 0.5 also applied to all thermally denuded data? Evaporating part of the organic aerosol (and therefore changing the org/sulfate ratio) could change the collection efficiency, which would bias MFR measurements. The authors could partially address this issue by comparing total SMPS and ACSM measurements (mass), i.e. the ACMS/SMPS ratio in the bypass and after the thermodenuder.

**AR2.5.** Yes, we have analyzed both bypass and heated (TD) ACSM data using a constant collection efficiency (CE) of 0.5. A comparison of ACSM/SMPS ratio in the bypass (slope = 0.95±0.006) and after the TD (slope = 0.91±0.009) data is shown in panels c and d in Fig S1. A slightly lower slope for the heated measurements suggest that CE could be ~4-8 % lower for the aerosol that passes through the TD. We did not attempt to use a different CE value for the thermodenuded aerosol based on this analysis because the SMPS measurement also could be affected by a potential change in particle morphology and shape upon heating. However, the bottom line is that the potential uncertainty that may arise from using a constant CE would be very low compared to the overall observed variability in measurements.

**R2.6.** Fig S7 seems to assume that all nitrate measured in the ACSM is inorganic. This seems to contradict discussion earlier in the manuscript of the potential importance of organic nitrates.

Previous work has shown that the ratio of NO+ to NO2+ fragments measured by AMS or ACSM instruments is quite different for organic nitrates and ammonium nitrate, and that the ratio can be used to estimate the fraction of measured nitrate due to organic nitrates. I suggest the authors use these measurements from the ACSM to estimate how much of the measured nitrate is organic vs. inorganic. Assuming full neutralization of sulfate by ammonium (which is reasonable in the presence of ammonium nitrate) could also be used to calculate the nitrate attributable to ammonium nitrate and, therefore, the nitrate due to organic nitrates.

**AR2.6.** The Nitrate measured by an ACSM would be a combination of both organic and inorganic nitrate. We changed the axis label for panel (b and d) of Fig.S7 from 'inorganic' to 'SO$_4$, NO$_3$, NH$_4$'. We explored the NO+ to NO2+ ratio from our ACSM measurements as suggested by the reviewer. A summary is given below:

| Measurements | NO+ to NO2+ ratio (campaign average ± 1 SD) |
|---|---|
| Centreville | 10.02 ± 3.47 |
| Raleigh | 5.93 ± 1.96 |
| Pure ammonium nitrate  (Cal aerosol) | 3.58 ± 0.84 |

The NO+: NO2+ ratios in both campaigns were significantly higher than that of pure ammonium nitrate aerosol in our instrument. This is consistent with a substantial contribution from organic nitrate and with our discussion of the potential importance of organic nitrates. However, an absolute quantification of organic nitrate is not critical for our analysis, as our paper focuses on the volatility of bulk OA as identified by the ACSM/AMS.

**R2.7.** Page 4 Line 12: "All Rts reported here are calculated assuming plug flow at room temperature" This implies that a plug flow (as opposed to parabolic) velocity profile is assumed in the evaporation model. Is this correct? It was suggested by (Cappa, 2010) that assuming plug flow profile would lead to underestimation of dHavp and overestimation of Csat. Have the authors explored different assumptions for gas velocity profile in the model?

**A2.7.** The Cappa (2010) paper discussed the effect of the laminar vs. plug flow assumptions on the derived saturation vapor pressure and enthalpy of vaporization of a single component aerosol.

The author notes that the effect of this assumption changes with the compound saturation vapor pressure. It is not clear what effect this assumption would have if a multi-component aerosol is considered. It should be noticed that a plug flow approximation is widely used (Lee et al., 2010; Riipinen et al., 2010; Saleh et al., 2008), with the residence time and non-uniform temperature effects on derived quantities being relatively small (Park et al., 2013). Based on the sensitivity analysis result presented in our earlier dual-TD method characterization paper (Saha et al., 2015) and a detailed two-dimensional laminar flow modeling effort in Park et al. (2013), this plug-flow assumption has relatively smaller influences on evaporation in a TD relative to values of ($C^*$, $\Delta H_{vap}$, and $\gamma_e$).

Minor comments:

**R2.8.** Page 5 Line 2: Please expand on what "instrumental inter-calibration factors" entail

**AR2.8.** To get directly comparable SMPS concentration data from 3 SMPSs running in parallel with our dual TD system, we ran them periodically in parallel to determine an inter-calibration factor. The inter-calibration factor is determined from a scatter plot of SMPS inter-comparison data collected by running 3 SMPSs in parallel on the bypass line. Among the 3 SMPS, we chose one as a reference upon which all corrections were based. The reference SMPS system was selected based on which yielded counts most consistent with the median of those measured during a group SMPS inter-comparison test of 8 systems from different laboratories during the SOAS field campaign. We added this discussion briefly in the paper:

*"To get directly comparable SMPS concentration data from 3 SMPSs running in parallel with our dual TD system, we ran them periodically in parallel on the bypass line to determine inter-calibration factors. Further details on SMPS inter-comparison are discussed in Saha et al (2015)."*

**R2.9.** Page 6 Line 23-24: Please clarify on how variability is calculated. Is it based on measured MFR values or OA measurements? Is it calculated over small intervals or over the entire course of campaign?

**AR2.9.** We added this discussion to the paper: *"Variability is based on measured MFR data. Raw data at each (T, Rt) condition were averaged over 20-30 minutes. At given TD operating*

*conditions (T, Rt), we defined ±1 standard deviation of MFR data (20-30 minute resolution) from the whole campaign as an indicator of observed variability."*

**R2.10.** Page 8 Line 31: Please provide references for previous field studies.

**AR2.10.** Page 9; Line 12-13: References to field studies ("Häkkinen et al., 2012; Huffman et al., 2009; Lee et al., 2010; Paciga et al., 2015; Xu et al., 2016") are added.

**R2.11.** Page 10 Line 8-10: Please add discussion on why observed slope may be different from the empirical relation determined by Epstein et al.

**AR2.11.** The Epstein et al. correlation was determined from range of compounds with known ΔHvap. However, it has been found that for this and other complex OA systems, a correlation other than the Epstein correlation better explains observations. For example, Ranjan et al.(2012) reported $dH_{vap}$ = 85-11logC* for gas–particle partitioning of POA emissions from diesel engine; May et al.(2013) reported $dH_{vap}$ = 85-4logC* for biomass burning POA emission. A key point is that like results from Ranjan et al. (2012), May et al. (2013) and many others, our estimated $dH_{vap}$ correlation for ambient OA is an empirical estimate, which explain our observations better than the Epstein correlation. We added this discussion in the paper. Page10, Line 28-33:

*"The Epstein et al. correlation was determined from range of compounds with known $\Delta H_{vap}$. Several recent studies of complex OA systems (May et al., 2013; Ranjan et al., 2012) have found that a correlation other than that from Epstein et al. better explains observations. For example, Ranjan et al.(2012) reported $\Delta H_{vap}$ = 85-11logC* for gas–particle partitioning of POA emissions from a diesel engine; May et al.(2013) reported $\Delta H_{vap}$ = 85-4logC* for biomass burning POA emissions. Similar to these and other studies, our $\Delta H_{vap}$ correlation for ambient OA is an empirical estimate which best explains our observations."*

**R2.12.** Page 2 line 6: ": : :SOA is formed in the atmosphere via condensation of low volatility products: : :" This statement is not inclusive enough. Please revise.

**AR2.12.** We revised the text. Page 2; Line 5-6: *"Secondary OA (SOA) is formed in the atmosphere via oxidation reactions of gas-phase organic species; it may also be formed by reactions in the particle (condensed) phase."*

Where, D is the diffusion coefficient, F is the Fuchs and Sutugin correction factor, $N_i$ is the number concentration of particles in size bin of $d_{p,i}$

**S3. Supplementary Figures**

[revised manuscript text omitted]

10 Centreville campaign.

[Figure]

**Figure S14:** Similar to figure 8 in the main text showing analysis results for Raleigh data set. Scatter plot of mean C* verses (a) rough HOA fraction, and (b) rough OOA fraction in total OA concentration during the Raleigh campaign. For rough HOA and OOA estimation method, see Fig S11 caption.

**S4. Supplementary Tables**

**Table S1:** TD kinetic model input parameters

| Parameters | Value | Notes |
|---|---|---|
| Density (kg m$^{-3}$) | 1400 | Kuwata et al., 2012 parameterization |
| Diffusion coefficient (m$^2$ s$^{-1}$) | 3.5 E-06 | Cappa and Jimenez (2010) |
| Surface tension (J m$^{-2}$) | 0.08 | Approximated as Pimelic acid, Bilde et al.(2003) |
| Molecular weight (MW) | $MW_i$ (g mol$^{-1}$) =169-28 ($log_{10}C^*_i$) | Approximated from Di-carboxylic acid |

5  **Table S2:** Statistical correlation (Pearson R value) between isoprene-OA fraction to $C_{OA}$ and $f_i$'s in any particular $C^*$ bin

| $log_{10}C^*$ bin | -4 | -3 | -2 | -1 | 0 | 10 | mean $C^*$ | $C^*_{eff}$ |
|---|---|---|---|---|---|---|---|---|
| Pearson R value | 0.02 | 0.29 | -0.07 | -0.06 | -0.14 | 0.04 | -0.06 | 0.19 |